## Registered report 

psychology

self-help plus, anxiety, PTSD, randomized controlled trial, nursing homes, care homes

**Author for correspondence:**
Elena Rusconi
e-mail: elena.rusconi@unitn.it

# Effectiveness of self-help plus (SH+) in reducing anxiety and post-traumatic symptomatology among care home workers during the COVID-19 pandemic: a randomized controlled trial

Marianna Riello[1,2], Marianna Purgato[3,4], Chiara Bove[1], Federico Tedeschi[3,4], David MacTaggart[5], Corrado Barbui[3,4] and Elena Rusconi[1]

[1]Department of Psychology and Cognitive Science, University of Trento, Corso Bettini, 31, 3868 Rovereto, Trentino, Italy
[2]Gruppo SPES, Trento, Italy
[3]WHO Collaborating Centre for Research and Training in Mental Health and Service Evaluation, Department of Neuroscience, Biomedicine and Movement Sciences
[4]Section of Psychiatry, University of Verona, Verona, Italy
[5]School of Mathematics and Statistics, University of Glasgow, Glasgow, UK

DM, 0000-0003-2297-9312; ER, 0000-0003-2700-205X

This article describes a randomized controlled trial to evaluate the effectiveness of a supervised online delivery of self-help plus (SH+), during the second wave of COVID-19 contagions in Northern Italy. The SH+ is a psychological intervention developed by the World Health Organization to increase a person's ability to deal with stress. In this trial, it was tested primarily as a tool to reduce anxiety and post-traumatic symptomatology in workers of residential nursing and care homes. In order to partial out non-specific effects of the intervention, the SH+ was compared to an equally supervised and structured alternative activity. Secondarily, in view of future emergencies, the potential of SH+ as a tool to reduce perceived stress, increase subjective well-being and foster individual resilience was explored. At post-intervention, the preregistered analysis revealed no difference in self-reported anxiety and/or post-traumatic symptomatology between the group receiving the SH+ and the group engaged in an

alternative activity. Some specific and positive effects of the SH+ intervention were only found on self-reported intervention effectiveness and engagement in exploratory analyses. These findings raise the question whether the previously documented effectiveness of the SH+ on self-reported symptomatology and on the prevention of psychiatric conditions could be attributed mostly to non-specific rather than specific factors connected with participant enrolment in a psychological intervention. Indeed, the effects of the SH+ had been previously compared only to the effects of not being engaged in any alternative activity (often described in the literature as 'treatment as usual'—or 'enhanced treatment as usual', when some relevant information is given to the control group as a one-off). Given the negative findings of this study, before the SH+ is implemented in clinical practice, further studies should be conducted to examine its short- and long-term beneficial effects, by means of randomized studies that employ alternative but similarly structured interventions as control conditions, aiming to minimize the confounding effect of non-specific factors.

# 1. Introduction

## 1.1. Background and rationale

As of 28 February 2021, the number of confirmed cases of COVID-19 has risen to about 113 million worldwide, including more than 2.5 million deaths [1]. The effects of the pandemic on mental health in the general population, and particularly in front-line healthcare workers, can be likened to those of a humanitarian emergency, given the sudden exposure of large numbers of people to multiple potentially traumatic events. Indeed, exceptionally high levels of psychiatric symptomatology have been reported around the world, in the general population and especially among healthcare workers, throughout the past 10 months [2–4].

In a previous study, we highlighted how nursing and care homes (NCHs) have experienced major challenges during the current pandemic, probably due to an unfortunate combination of factors [5]. We also reported a higher than expected frequency of anxiety and post-traumatic symptomatology in a large sample of workers from Northern Italian NCH—particularly in women and those who had been in contact with COVID-19-positive individuals—at the end of the first wave of COVID-19 contagions. Although it is not possible to establish a causal relationship between this situation and the current pandemic based on the evidence, or to know to what extent the evidence captures an issue that predates the pandemic, we suspect that the sudden increase of multiple potential stressors brought about by the pandemic played a non-negligible role. Indeed, recent or prolonged exposure to intensely stressful situations is a well-documented risk factor for the development of both generalized anxiety and post-traumatic symptomatology [6]. Moreover, because the pandemic has not yet receded and we are now in the middle of a second wave of contagions, the current situation could further exacerbate the issue and psychological support to workers who assist vulnerable individuals on a daily basis should be considered as a priority [7]. However, in consideration of the large scale of the problem on the one hand and the current restrictions to social contact on the other, it would not be possible to provide traditional one-to-one or tailored group assistance in person, and in a timely way. Furthermore, due to the limited availability of vaccines (especially in the poorest parts of the World), contrasting public attitudes about vaccination and the very recent spread of new and more infectious variants of the virus, such restrictions seem unlikely to be lifted any time soon. In Italy, for example, the state of emergency has been extended to 31 December 2021 [8] but further extensions are expected. Thus, alternative and wide-reaching online interventions are particularly suited to these days and times.

In this randomized controlled trial (RCT), we tested the efficacy of a self-help tool (self-help plus, SH+) recently devised by the World Health Organization (WHO) [9]. The SH+ is a low-intensity psychological intervention developed by the WHO and collaborators working in the field of humanitarian emergencies. It has been developed with populations that are numerous and/or hard to reach by professionals, with a view to make 'mental healthcare more widely available to those in need'. 'It is intended to be transdiagnostic' and 'both meaningful and safe for people with and without mental disorders' [9, p. 295]. The SH+ appears particularly useful within a scalable intervention approach, as an initial move to prevent the development of psychological disorders [10]. The SH+ was originally devised for administration in a group setting [9] but is now also disseminated as *Doing What Matters in Time of Stress*, an SH+ version consisting of an illustrated guide with audio files for individual self-help work

[11]. It was used here as a first-step intervention to reduce the documented frequency of post-traumatic symptomatology and anxiety in individuals working in NCH during the COVID-19 pandemic. Both generalized anxiety and post-traumatic symptomatology can be significantly reduced by stress-relief techniques [12,13] and, for preregistration purposes, they were considered here in a combined index, in accordance with both the transdiagnostic character of the SH+ and with the general applicative scope of our project. Indeed, our primary aim was an evaluation of the effectiveness of SH+ in reducing, over and above an alternative intervention, the overall frequency of psychological symptoms in a volunteer sample of workers from NCH. In order to partial out non-specific effects of the intervention, the comparison group consisted of an equally supervised and structured control activity. Additionally, we explored the effects of SH+ on the two conditions separately in further analyses. Since SH+ is a relatively light and short-lasting intervention, we focused on its effects immediately post-intervention (though we also included a test for its medium-term efficacy) and we expected its utility to be maximal in emergency situations. Given the agility of the intervention, it could be repeated as necessary in other emergency situations if beneficial effects were found in the short term.

The potential beneficial effects of SH+ on psychological symptoms may be related to its main core components. The intervention—based on acceptance and commitment therapy (ACT)—aims to enhance psychological flexibility and improve coping strategies to deal with adversity [14,15]. SH+ encourages participants to respond more adaptively to fluctuating situational demands, by finding ways of acting in accordance with their values, even in the face of difficult external circumstances. Participants could learn to accommodate difficult thoughts and feelings, becoming flexible in the management of adversity through the use of mindfulness techniques. This might help improving coping strategies, thus reducing the levels of anxiety and PTSD symptoms. The goal of ACT is to assist individuals to engage in behaviours that work best in allowing them to reach their stated goals, and the reduction of anxiety and/or PTSD symptomatology may thus be regarded as a by-product of this intervention aim [15,16]. Accordingly, ACT might lead to broader substantial changes than other forms of psychological interventions regarding psychological functioning [17].

The SH+ was recently tested in a large cluster RCT with the aim of reducing psychological distress and improving functioning at a three-month follow-up in a population group of 694 severely distressed South Sudanese female refugees in Uganda. As planned in the present trial, the follow-up assessments of the trial in Uganda were conducted immediately post-intervention, and after three months. Findings from the trial in Uganda are promising, highlighting a significantly greater reduction in psychological distress measured with the Kessler-6 (K6) rating scale after intervention ($\beta$ −3.25, 95% confidence interval (CI) −4.31 to −2.19; $p < 0.0001$; $d$ −0.72) and after three months relative to the enhanced usual care ($\beta$ −1.20, 95% CI −2.33 to −0.08; $p = 0.04$; $d$ −0.26). In addition to the decrease in distress, the authors identified a significant improvement of PTSD symptoms at post-intervention ($\beta$ −3.53, 95% CI −4.67 to −2.38; $p < 0.0001$; $d$ −0.68) and at three months ($\beta$ −1.55, 95% CI −2.87 to −0.24; $p = 0.02$; $d$ −0.30) [18].

That said, these findings needed to be replicated in our target population group as they were produced in a different population group (refugees), in a different setting (refugee camp in a low-income country; [18]) and using a face-to-face group delivery modality. Even though there are some commonalities in the choice of rating scales for measuring trial outcomes (i.e. WHO-5), the population group of the Ugandan trial (mainly composed of severely distressed women), together with cultural peculiarities and a limited amount of resources led to the choice of different (still validated) outcomes and different instruments for measuring outcomes.

To test the effectiveness of the SH+ in its *Doing What Matters in Time of Stress* version, we put in place an online support structure for the participants, to help them work through all the materials individually and with a steady pace within a set time window of five weeks. Intervention effectiveness was measured as the frequency difference in anxiety and/or post-traumatic symptomatology at the end of the intervention window between a group of participants engaged in the SH+ and a group of participants engaged in an alternative activity. The alternative activity was based on reading and reflecting on a short novel from an illustrated booklet with associated audio files. The alternative activity was designed to (i) partial out improvements that could be elicited by assistant attention, individual engagement and expectations (all known contributors to the positive response to any therapeutic act, e.g. [19,20]) rather than by specific content, (ii) comply with the request, from both trade union representatives and the consulted NCH, to conduct a study from which all volunteering workers could be expected to receive at least a minimal benefit [19–22]. Therefore, a similar amount of time and commitment was requested of the participants in that group, a similar amount of support was given to them as that given to participants in the SH+ group and both groups received materials that

could be helpful and easily related to their personal situation. On the other hand, although some benefit from the assigned reading and guided reflection tasks could not be excluded [21,22], these materials were not expected to provide benefits as broad and systematic as those provided from the ACT/mindfulness-based tasks and materials given to the SH+ group [16].

If the intervention was shown to be effective with our sample of participants, its timely deployment on a larger scale could be recommended. To maximize the benefits, NCH could also consider offering it as a possible training activity during working hours, thus potentially enrolling a large proportion of their workers. Indeed, this relatively low-cost intervention could come with a favourable cost–benefit trade-off, when compared to tailored psychological assistance for a vast proportion of workers in severe distress during or soon after the pandemic. The format and support structure adopted in this study, which can be easily replicated, could enable a fast and efficient roll-out.

The study also included a series of secondary outcomes, which helped us gather a more comprehensive picture of the possible effects of SH+ and identify directions for future studies. As mentioned before, we were able to explore the effects of SH+ on anxiety and post-traumatic symptomatology separately. Additionally, we considered the attrition rate throughout the intervention as an indicator of intervention acceptability. We also performed a further follow-up 14 weeks after the intervention for both groups. The frequency difference in anxiety and/or post-traumatic symptomatology between the SH+ group and the alternative activity group at the 14-week follow-up could be an indicator of medium-term effectiveness of the SH+ intervention. In the view of the possibility of repeated waves of contagions and of similar future emergencies, we consider it important to assess and identify interventions that can boost individual resilience. Therefore, a self-report measure of individual resilience was also included in our protocol, along with measures of perceived stress and subjective well-being. If SH+ improved such measures, its potential to serve also as a proactive intervention to boost resilience, increase well-being and decrease baseline perceived stress once the current pandemic has receded, and thus decrease the probability to develop psychological symptoms in a future emergency [23,24], should be further investigated.

## 1.2. Aims and objectives

In summary, the overarching aim of this RCT is to evaluate the effectiveness of SH+ as a psychological intervention to increase the ability of NCH workers to cope with stressful situations during the pandemic. This was operationalized as a reduction, at the end of the intervention, in the frequency of anxiety and/or post-traumatic symptomatology in the SH+ group compared to a group receiving an alternative but similarly structured intervention.

The secondary, exploratory, objectives of this study are: to probe the effects of SH+ on anxiety and on post-traumatic symptomatology separately, to probe intervention acceptability (represented by attrition rates) and its medium-term effectiveness at a 14-week follow-up, and to evaluate potential SH+ effects on individual resilience, perceived stress and well-being.

The hypothesis and analysis plan for the main outcome were preregistered with the protocol, in compliance with the Registered Report format requirements (see also table 1). However, also our planned exploratory analyses on the primary and secondary outcomes were pre-specified, in line with conventional trial reporting.

## 1.3. Main study hypothesis

The SH+ group will report a lower proportion of any level (from mild to severe) of anxiety (GAD-7 score ≥ 5) and/or[1] post-traumatic symptomatology (IES-R score ≥ 9), as measured with IES-R and GAD-7, respectively [5,25], at the end of the intervention window (one-week follow-up) compared to the control group.

---

[1]This means that, as in Riello et al.'s epidemiological study, a person satisfied the criterion in the following cases:

—s/he passed the threshold for IES-R but not the threshold for GAD-7;

—s/he passed the threshold for GAD-7 but not for IES-R;

—s/he passes the threshold for IES-R and the threshold for GAD-7. In other words, an individual passing both the IES-R threshold and the GAD-7 threshold still counted as 1 in our frequency calculation.

**Table 1.** Summary of our preregistered research question and hypothesis, sampling, analysis and interpretation of possible results.

| | |
|---|---|
| question | Will the group of care home workers receiving SH+ report a lower proportion of anxiety and/or post-traumatic symptomatology by the end of the intervention, compared to the group engaged in an alternative activity? |
| hypothesis | The proportion of individuals reporting anxiety (GAD-7 score $\geq$ 5) and/or post-traumatic (IES-R score $\geq$ 9) symptomatology will be lower in the SH+ group than in the alternative activity group at the one-week post-intervention follow-up. |
| sampling | Our minimum sample size will be of 89 participants per group, after exclusions for attrition and non-compliance. This will ensure a power of at least 90% for a unidirectional test between the two proportions of participants with self-reported symptoms at a 0.05 level of significance. Considering expected attrition/non-compliance rates, our minimum recruitment target will be of 105 participants per group (see §2.1.2). |
| analysis | Participants will be deemed to report mild-to-severe symptoms if their scores surpass the threshold in at least one of the two scales (i.e. obtaining a score greater than or equal to 5 for the GAD-7 and/or greater than or equal to 9 for the IES-R). The proportion of participants reporting mild-to-severe post-traumatic and/or anxiety symptoms at one-week follow-up will be compared between the SH+ and the alternative activity group using a one-tailed $\chi^2$-test with significance level set at 0.05. |
| interpretation | *Case 1*<br><br>$p \geq 0.05$: SH+ is not more effective than the alternative activity for lowering the proportion of care home workers reporting anxiety and/or post-traumatic symptomatology by the end of the intervention period.<br><br>*Case 2*<br><br>$p < 0.05$: SH+ is more effective than the alternative activity for lowering the proportion of care home workers reporting anxiety and/or post-traumatic symptomatology by the end of the intervention period. |

# 2. Material and methods

This is a single-blind, prospective, randomized, parallel-group study that followed participants over a period of 20[5] weeks, from 10 March 2021 to 3 August 2021 (Week 0: consent, baseline; Week 1: randomization; Week 2–Week 6: intervention; Week 7: post-test 1; Week 20: post-test 2). The trial was approved from the Internal Review Board of Gruppo Servizi Pastorali Educativi Sociali (SPES) and started after receipt of in-principle acceptance from Royal Society Open Science. Figure 1 provides a flow-chart of the RCT process.

## 2.1. Participants

### 2.1.1. Sample size and power calculations

Previous studies have reported the stability over a year of the prevalence for psychiatric symptomatology in connection with a direct experience of the SARS pandemic, even though the pandemic had by then receded [26,27]. There is no comparable evidence for COVID-19, given that the state of emergency is still ongoing. However, there is preliminary evidence suggesting a progressive worsening of healthcare workers' mental health during the pandemic, compared to non-healthcare workers and the persistence of higher levels of psychological distress in front-line healthcare workers even after the peak of contagions [28,29].

Based on recent prevalence figures, we expected about 85% of our workers in NCH to report mild-to-severe symptoms of post-traumatic symptomatology and/or anxiety (this derives from data collected by Riello *et al.* [5] between June and July 2020; more precisely, 42% of the workers were expected to report mild symptoms and 43% moderate-to-severe symptoms). Because we were in the middle of an evolving

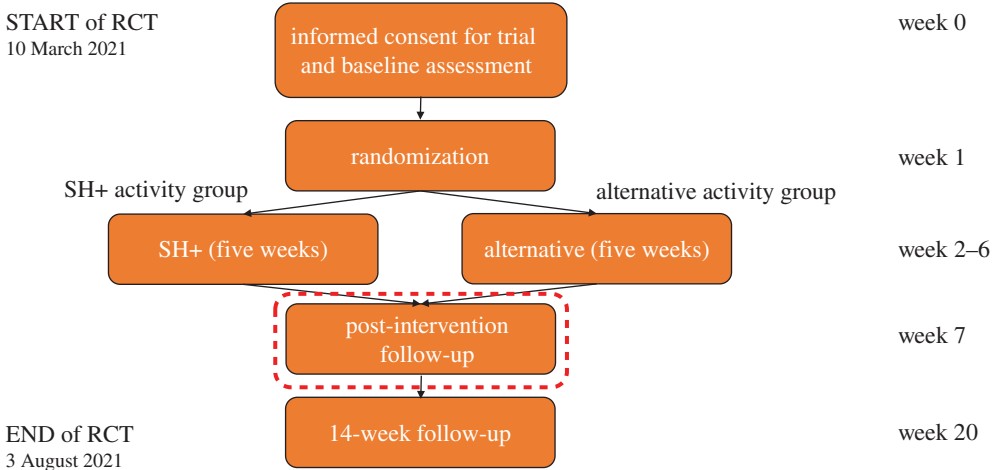

**Figure 1.** RCT flow diagram. The red dotted rectangle signposts the primary outcome data lock. The trial took 20, instead of the expected 18, weeks due to a two-week delay in the start of post-test 2. This delay was necessary to maintain parallel testing across NCH during the holiday season (see footnote 2).

situation (a second wave of COVID-19 contagions, the initial limited availability of vaccines, the appearance of new and more infectious variants of the virus, etc.), we reasoned that the current prevalence could actually be higher than the one based on data collected after the first wave of contagions had receded. In any case, both participants in the experimental group and participants in the control group were expected to be equally affected by the situation.

For the purpose of this power analysis, we assumed a prevalence, in the absence of any intervention, of 85% (corresponding to that recorded from the same population after the first wave of contagions and well before the start of a second wave). To account for a possible protective effect of the alternative activity, we hypothesized a risk for the control group of 80%. Also, we considered an absolute risk reduction relatively to the alternative activity of 20% (corresponding to a risk ratio, RR, of 0.75 and leading to a 60% prevalence) in the proportion of self-reported low-to-severe symptoms of anxiety and/or post-traumatic symptomatology in the SH+ group at one-week follow-up to be the minimum clinically relevant effect, and we considered the CI to be reasonably precise by having an upper limit of the CI within 25% of an estimated value of risk ratio. The number of participants necessary to achieve a power of at least 90% for a one-sided test between the two proportions of participants with self-reported symptoms at the one-week follow-up at a 0.05 level of significance, and for the upper limit of the 95% two-sided CI to be within 25% of estimated value of RR, is 89 per group, after exclusions due to attrition or non-compliance. We calibrated our recruitment efforts accordingly.

Since we could not provide an accurate prediction of the attrition/non-compliance rate with this population (in our previous epidemiological study, the rate of attrition at the single data collection timepoint was 5% but the current study includes two essential time points for data collection: baseline and one-week post-intervention assessment), we assumed an overall 15% rate of attrition and/or non-compliance, also based on similar literature (e.g. [30]), and set a minimum sample size of 105 participants per group as our recruitment target. The script of our sample size and power calculations is available on the Open Science Framework (OSF) [31].

### 2.1.2. Recruitment and RCT inclusion criteria

NCH were recruited via convenience and snowball sampling from the list of centres that had previously participated in our prevalence study, and also other contacts. Twelve centres had already agreed to participate at the time of our Stage 1 submission and we continued recruiting until Stage 1 approval.

Soon after receipt of Stage 1 approval, we commenced workers' recruitment. All workers in participating NCH were contacted at once via mailing lists and informed about our intervention study, which would take place over five weeks plus three months[2] from the start of the intervention. Those meeting the inclusion criteria outlined below, completing an initial survey (demographic

---

[2]Overall, the trial took 20, instead of the expected 18, weeks (see also figure 1) due to a two-week delay in the start of post-test 2. This delay was necessary to preserve parallel testing across NCH during the summer holiday season. At recruitment, participants were

information and baseline assessment) and passing the validity check at the baseline assessment (see §2.4) were then randomly assigned to one of the two activity groups. Randomization was stratified by centre and the RCT was conducted in accordance with the Consolidated Standards of Reporting Trials statement [32,33].

Inclusion criteria (for randomization purposes) were the following:
(a) Providing online informed consent.
(b) Aged 18 or above.
(c) Able to speak, read and understand Italian fluently (self-assessed).
(d) Works in a Northern Italian[3] nursing or care home as medical/healthcare, administrative or technical staff.[4]
(e) Passes the validity check at the baseline assessment (see details below).

### 2.1.3. Randomization

Randomization occurred at the individual level and was stratified by recruiting centre. To avoid contamination, only one person per household was randomized. Randomization was centralized and coordinated by a researcher (the website manager) different from the participant recruiter, the assistant, the data analyst and the data manager. This researcher did not have access to baseline assessment outcomes. All eligible participants were randomly assigned to one of the two groups with an equal probability of assignment to each group (allocation ratio 1 : 1). The randomization schedule was generated using Stata 16 [34] and the randomization list was accessible only to the website manager for the assignment of activities to individual participants. After random allocation, a unique identification number was assigned to each participant. In accordance with the Declaration of Helsinki, participants' confidentiality was preserved at all times. Further, potentially identifying contents of the baseline and follow-up forms were not disclosed to any third party; individual data have been shared with the scientific community on the OSF [31] after elimination of any sensitive information that could lead to recruiting centres' or participants' identification or geo-localization.

## 2.2. Characteristics of the intervention

After providing informed consent, filling a demographic questionnaire and completing a baseline assessment, participants were randomly assigned to two intervention groups. The informed consent form specified that this intervention study comprised different types of activities (teaching a series of psychological techniques to cope with stress or promoting reflection on relevant topics such as work roles and related emotions), that volunteers would be randomly assigned to a given set of activities and that we expected beneficial effects from all sets of activities. In other words, participants received accurate information, while remaining blind to our main study hypothesis. This approach aimed to minimize contamination between groups and the presence of differential expectations or motivational states once the activities were disclosed to individual participants in Week 2 of the RCT.

At the beginning of the study, all participants were also given information about available mental health services and community networks that provide free support for workers in times of COVID-19 at a national level. At the two data locks following the intervention (i.e. at the one-week follow-up and 14-week follow-up; see figure 1), participants were requested to flag whether they had made use of any of these services (or other professional psychological/psychiatric support and/or psychoactive drugs) since the beginning of the study.

---

informed the study would last about five weeks and a further three months from the start of the intervention (see Materials available on the OSF); no exact dates were communicated.

[3]The North of Italy includes the following eight areas: Emilia Romagna, Friuli Venezia Giulia, Liguria, Lombardia, Piemonte, Trentino Alto Adige, Valle d'Aosta and Veneto.

[4]Administrative staff included: directors, nursing coordinators, administrative collaborators or assistants and secretaries; Medical/healthcare staff will include: physicians, nurses, healthcare auxiliary staff, physiotherapists, experts in psychiatric rehabilitation, speech therapists and psychologists; Technical staff will include: educators, entertainers, mediators, caseworkers, trainers, sociologists, specialized auxiliaries, technicians for the maintenance of the building and cleaning staff. These are the same three categories that contributed to the prevalence study of anxiety and post-traumatic symptomatology in NCH conducted by Riello *et al*. [5] after the first wave of contagions.

In the remainder of this section we describe the rationale and structure of the activities that were proposed to the two study groups: the SH+ intervention and the alternative activity (control). The materials have been made available on the OSF [31].

### 2.2.1. SH+

The characteristics of the SH+ intervention, especially in its most recent version as *Doing what matters in times of stress* [11], an individual self-help audio-visual tool, are ideally suited to overcoming the restrictions to interpersonal contact in an emergency situation such as the current pandemic. Indeed, SH+ includes a pre-recorded audio course, originally meant to be delivered by facilitators in a group setting over five 2 h sessions [9,10,18] and an illustrated self-help book. There is some redundancy between the pre-recorded audio course and the self-help book, so the self-help book could be used by itself. However, guided self-help has been found to be a more effective method of delivery compared to 'pure' self-help, with effects that are comparable to face-to-face psychological treatment [35]. In the SH+ version *Doing what matters in times of stress*, devised for individual use, the audio files are considerably shorter, thus making the intervention even more agile [11].

The current COVID-19 restrictions would make in-presence meetings impracticable and these workers' busy schedules and variable access to technological devices would represent an obstacle to organizing online group sessions. Therefore, we used the *Doing what matters* materials for supervised self-help activities and instructed individual participants to listen to sections of the audio course and go through the corresponding sections of the book autonomously, in moments of their choosing, within a pre-specified time window. Access to the materials was possible at any time of the day and from individual mobile phones via a dedicated website. At the end of each session, participants were able to log their activities, by selecting pre-compiled descriptors, and to fill in an online diary.

More precisely, participants in the SH+ group were given access to written and recorded materials for each of the five core component themes and activities of SH+ taken, with minimal adaptations,[5] from the official Italian translation of the *Doing What Matters in Time of Stress* version of SH+. In the first intervention week they received materials concerning 'Grounding', in the second 'Unhooking', in the third 'Acting on your values', in the fourth 'Being kind' and in the fifth 'Making room'. These materials can be found on the OSF [31]. They were required to access the material and work through it at their leisure during each corresponding week. They also received WhatsApp/text/email reminders—identical in number and content to the reminders sent to the control group—from the assistant, a professional-in-training, with a psychology background, who was supervised by academics and healthcare professionals throughout the intervention and the alternative activity delivery.

### 2.2.2. Alternative activity (control)

Participants in the control group were engaged in an alternative activity, which was built around the reading of the Italian translation of a short novel, *Leaf by Niggle* [36]. Being a lesser known work of a non-Italian modern writer, *Leaf by Niggle* was unlikely to be already familiar to any of our participants. It did not teach mental techniques to achieve a better management of stress in any difficult situation but likely provided compassionate reading through relatable characters. It was not written, nor was it expected, to exert a significant and direct impact on an individual's ability to cope with stressful situations or to provide a resilience toolkit. If effective in any way on people's mood and ability to cope with stress, its effect should be more transient and variable between individuals than that of the SH+ intervention, which explicitly provides tools to act on common symptoms of psychological distress in a systematic way [9].

There was a set of *ad hoc* written materials, *ad hoc* audio files and exercises to match the individual activities requested by the SH+. Audio files and exercises engaged participants in a generic reflection on the contents of the story, linking them to the vicissitudes of other fictional or historical characters, or relating them to their own work. Participants in the control group were given access to their

---

[5]The required adaptations consist of the following:

—the booklet and the audio-files in the *Doing What Matters* format of SH+ were not released in their entirety to participants but were made available with a paced release; as a consequence, the initial instructions differed from the original;

—the summary page of each part that can be found at the very end in the original booklet was moved at the end of the materials for the corresponding part, which was released on a weekly schedule;

—the audio files were released on a weekly basis with their corresponding materials.

**Table 2.** Main outcome measures and respective time points. After an initial baseline data lock, participants were assessed for all outcomes at the end of the intervention (one-week follow-up) and at a 14-week follow-up (figure 1). The primary outcome of the study was the number of participants passing the threshold for mild symptoms at IES-R ($\geq$9) and/or GAD-7 ($\geq$5) at the end of the intervention.

| concept | baseline | post-intervention follow-up | 14-week follow-up |
|---|---|---|---|
| post-traumatic symptomatology | IES-R | IES-R | IES-R |
| anxiety symptoms | GAD-7 | GAD-7 | GAD-7 |
| demographics | enrolment form | — | — |
| dropout | — | n/N | n/N |
| resilience | CD-RISC 25 | CD-RISC 25 | CD RISC 25 |
| subjective well-being | WHO-5 | WHO-5 | WHO-5 |
| | well-being index | well-being index | well-being index |
| perceived stress | PSS 10 | PSS 10 | PSS 10 |

material via the same dedicated website as participants in the SH+ group. In each intervention week they received new materials to progress with the story, in parallel with the intervention group. More specifically, they received materials under the following titles: 'Kind heart' in the first intervention week, 'Altruism' in the second, 'Gratitude' in the third, 'Collaboration' in the fourth and 'What matters' in the fifth. These materials can be found on the OSF [31]. Participants were required to access the materials and work through them at their leisure during each corresponding week. During each week, they also received WhatsApp/text/email reminders—identical in number and content to the reminders sent to the SH+ group—from the same assistant as the SH+ group. Materials for the control group were created *ad hoc* by the researchers, using published materials and adapting them to closely match the format and time requirements of the SH+ materials.

## 2.3. Primary and secondary outcomes

Outcome measures and collection time points are detailed in table 2. The primary outcome of this study is the proportion of any level (from mild to severe) of self-reported anxiety and/or post-traumatic symptomatology as measured at one-week follow-up by GAD-7 (cut-off score: 5; [5,25]) and IES-R (cut-off score: 9; [5,25]), respectively. The other outcome measures shown in table 2 have been included in the protocol for exploratory purposes. All the measures can be viewed on the OSF [31].

### 2.3.1. Post-traumatic symptomatology

For comparability with our previous study on the same population and other relevant COVID-19 literature ([5], e.g. [25]), the Impact of Event Scale-Revised (IES-R; [37]) was used to collect non-diagnostic information on post-traumatic symptomatology in emergency situations. This scale is one of the most widely used measures to assess trauma dimensions, and has shown good test–retest reliability and psychometric properties across languages and ranges of populations exposed to potentially traumatic events (e.g. [37–40]). It assesses post-traumatic symptomatology with reference to the last seven days, using 22 5-point Likert-scale items with responses ranging from 0 (not at all), 1 (a little bit), 2 (moderately), 3 (quite a bit), to 4 (extremely). Total scores range from 0 to 88. Only scales with all items completed were considered and entered in the study for statistical analysis. Total scores were interpreted as follows: normal (0–8), mild (9–25), moderate (26–43) and severe (44–88) post-traumatic symptomatology. Because the original IES-R instructions require respondents to refer to a specific stressful and potentially traumatic life event, our participants were instructed to respond with reference to any stressful event that happened to them at work in relation to the current pandemic (e.g. death of a patient or of a colleague, having to implement new and complex safety routines, etc.; see also [5]).

### 2.3.2. Anxiety symptoms

For comparability with our previous study on the same population and other relevant COVID-19 literature ([5], e.g. [25]), the 7-item GAD (GAD-7; [41,42]) was used to collect information on anxiety

symptoms. This scale is short to administer, has good psychometric properties and test–retest reliability both in psychiatric populations and in the general population across languages, and focuses on GAD symptoms—a disorder with a high degree of comorbidity, whose core feature (worry) can be found across a number of psychological disorders (e.g. [43–46]). It measures the frequency and severity of GAD symptoms with reference to the last two weeks, using seven 4-point Likert-scale items with responses ranging from 0 (not at all), 1 (several days), 2 (more than half the days) to 3 (nearly every day). Total scores range from 0 to 21. Only scales with all items completed were considered and entered in the study for statistical analysis. Total scores were interpreted as follows: normal (0–4), mild (5–9), moderate (10–14) and severe (15–21) anxiety.

### 2.3.3. Demographics

At enrolment we collected self-reported demographic data from respondents including: age (18–25, 26–30, 31–40, 21 41–50, 51–60, greater than or equal to 61), gender (male or female), education level (primary school, middle school, high school diploma or university degree), job title (administrative, healthcare, technical, other), information about the geographical location of a respondent's nursing/care home (to make sure that we could exclude all those outside of the northern region). We also asked respondents to indicate whether in the last two weeks they came directly in contact with COVID-19-positive colleagues or patients, as this was linked with higher odds of anxiety and/or post-traumatic symptomatology in our previous study [5].

### 2.3.4. Dropout

Attrition rates, calculated as number of participants not responding to the post-intervention assessment ($n$) over the total number of participants enrolled in each group ($N$), were taken as a measure of intervention acceptability.

### 2.3.5. Well-being

The WHO-5 Well-being Index is a short instrument measuring subjective quality of life based on positive mood, vitality and general interest. It has five items rated on a 6-point Likert scale with responses ranging from 0 (at no time), 1 (some of the time), 2 (less than half the time), 3 (more than half the time), 4 (most of the time) to 5 (all of the time). Total scores range from 0 to 25, 0 representing the worst possible and 25 representing the best possible quality of life. This scale is brief, non-invasive and has been found to have adequate validity across a wide range of languages and populations [47]. No cut-offs were applied to the scores of this scale for the purpose of the present study but they were considered as a continuous outcome.

### 2.3.6. Resilience

To measure personal resilience, the Connor–Davidson Resilience Scale 25 (CD-RISC 25) was used, which was originally developed and tested for sensitivity to treatment effects in clinical trials with GAD and PTSD patients, in addition to a variety of other populations [48]. It consists of 25 items that should be answered, with reference to the past month, and that are rated by a 5-point Likert scale with responses ranging from 0 (not true at all), 1 (rarely true), 2 (sometimes true), 3 (often true) to 4 (true nearly all the time). Total scores range from 0 to 100. It has sound psychometric properties, high test–retest reliability and is one of the best available resilience scales for both the general population and clinical populations [48,49]. No cut-offs were applied to this scale for the purpose of the present study but it was considered as a continuous outcome.

### 2.3.7. Perceived stress

The Perceived Stress Scale (PSS) is the most popular scale for the assessment of subjective stress. The respondent is asked to rate the degree to which s/he perceives his/her life as uncontrollable, unpredictable and overloading in the past month [50]. It is related to psychological stress and self-reported health (depressive and physical symptomatology; [51]), and correlates with biological markers of stress and disease [52]. Items are rated on a 5-point Likert scale, with responses ranging from 0 (never), 1 (almost never), 2 (sometimes), 3 (fairly often) to 4 (very often) and total scores range from 0 to 40. No cut-offs were applied to this scale for the purpose of the present study but it was considered as a continuous outcome.

## 2.4. Validity checks

To control for careless responding, we added a few extra questions (one at the end of each questionnaire) similar in spirit to those found in the Information section of the Italian Wechsler Memory scale [5,53]. These were very simple questions appended to the primary outcome measures (e.g. 'Who is the current Queen of the United Kingdom?', 'What is the capital of Italy?', 'What is the name of the current Pope?'). If a respondent from either the SH+ or the control group did not answer correctly to at least two of the three questions associated with the main questionnaires at the baseline assessment (demographic, IES-R and GAD-7), they would still be given individual access to one of the interventions but they would not be included in the preregistered randomization and data analysis. Note that, in order to avoid potential biases in participant exclusion related to the intervention received, which could jeopardize the benefits of randomization, we would exclude participants based on careless responding at the baseline assessment only. Indeed, a lack of careless responding at baseline (i.e. before randomization has taken place and thus before participants can access any website contents) was taken as a proxy for a lack of careless responding at the one-week post-intervention assessment. No exclusions would, therefore, be performed on the basis of participants' responses to the same control questions at the one-week post-intervention data lock for the purpose of the preregistered primary outcome analysis. It would still be possible, however, to exclude participants (if any) giving more than one inaccurate response to the control questions during any data locks in further analyses.

We also used IP/device tracking to detect and prevent duplicate responses (no more than one response could be sent from the same device). Participants were contacted and identified through their organization, requested to avoid environmental distraction when completing the questionnaires and to possibly use their personal rather than work devices (as an extra measure to preserve their privacy). This aimed to minimize collaborative answering or contamination between groups. Participants were requested not to share any information with their colleagues and to report whether they lived in the same household as colleagues working in the same or a different care home. We asked them to indicate their names. If more than one person from the same household agreed to participate in the RCT, we only included the first to join in the randomization.

We included at each data lock two control items relating to (1) reading habits for novels and short stories (i.e. 'How many novels and short stories do you normally read in a year?') and (2) appreciation of novels and short stories (i.e. 'How useful do you find reading novels and short stories?'). Individual responses to these items were either expected to remain unchanged across data locks or, in the case of item 2, they could change for the control group (due to its activity focus) but not for the SH+ group.

In order to monitor individual engagement with the intervention and use of materials, we collected data about the dates and times of log-ins to the website where materials were made available from week to week, and asked our participants to log their activities by choosing from a pre-compiled list of activities before logging-out and at the end of each intervention week. They were also given the possibility to write their own notes in a textbox. We included in our preregistered analysis all participants who completed the post-intervention assessment and logged-in to the website at least once, regardless of their recorded levels of engagement. That is, for our preregistered analysis, we adopted a conservative intention-to-treat approach [54].

A subjective measure of expectations was collected before randomization, to assess whether our initial information was effective at inducing similar expectation in all our participants. Perceived engagement, intervention effectiveness, commitment[6] and expectation fulfilment were also measured at the two post-intervention assessments, to check whether the proposed activities (SH+ and control) were perceived as similarly engaging and effective and whether they fulfilled our participants' expectations to a similar extent (please see materials available on the OSF [31]).

An assistant checked in regularly with the participants via WhatsApp, email and text messages, to monitor their progress and to encourage compliance. When measuring outcomes post-intervention, we included a question to check participants' perception of assistant helpfulness (On a scale from 1 to 10, how helpful has the assistant been to you?), which allowed us to test whether helpfulness perception was different between groups.

---

[6]As per approved protocol, the added variable (Q17 in the outcome measures file available on the OSF) was measured along with perceived effectiveness, assistant helpfulness, engagement and fulfilment of expectations. Its inclusion in our exploratory analysis was planned before data collection and the absence of its mention in the previous manuscript was an oversight.

## 2.5. Masking

The research assistant sending reminders to participants to access the online sessions of both the SH+ intervention and the control activity was masked. Moreover, both the data manager and the data analyst, performing all basic and in-depth analyses, were masked to the participants' allocation status. Only the web manager was aware of the allocation status of participants, as he had to give access, via an *ad hoc* website, to the assigned materials to each participant. On the other hand, he did not have access to baseline assessment outcomes, which were collected via a separate online system (SurveyMonkey®) and were accessible only to the data manager. Participants were masked to the aims of the study and the number of groups. They were primed at recruitment to develop positive expectations, whatever their assigned activities would be (see also §2.2).

## 2.6. Planned adverse event reporting management

Any serious adverse events reported spontaneously by the participants or observed by the research staff would be recorded on a specifically developed form. Data on the relationship with the study intervention, the action taken regarding intervention and the outcome of the adverse event would be collected. An event was considered a potential adverse reaction if it were an undesirable experience occurring to a participant during the study, whether or not it was considered to be related to the research procedure. This definition includes all aspects of mental health and psychological functioning, but also any other undesirable experiences. The Internal Review Board would review spontaneously reported serious adverse reactions (e.g. suicide attempts) within 48 h, if any, while general adverse reactions would be reviewed by the Internal Review Board in regular meetings.

## 2.7. Statistical analysis

### 2.7.1. General approach

The statistical analysis was masked. The data analyst was blinded to the intervention groups until completion of the analysis. Moreover, he was not involved in determining participants' eligibility, in administering the interventions, in measuring the outcomes or in entering the data. All analyses were performed using Stata 17 [55]. Three main data locks occurred during the study. The first occurred on week 0, at the beginning of the RCT, when baseline information was collected from all participants with the primary and secondary outcome measures; the second occurred on week 7, just after the end of the intervention period; the third occurred on week 20, at the 14-week follow-up.

Only the first two data locks (i.e. baseline and one-week post-intervention follow-up) and the primary outcome measures (i.e. IES-R and GAD-7) are relevant for the preregistered analysis. For every data lock, all questionnaire items were set as requiring a response before proceeding to submission. Therefore, as in [5], there were no missing data at the item level. However, there could be missing data at the outcome measure level. In case individual data were available for none or only one of the scales contributing to the primary outcome at the one-week post-intervention assessment, such participants would be counted among the dropouts (i.e. they would contribute to the attrition rate) for the main preregistered data analysis.

Before proceeding to the preregistered data analysis, only participants (if any) who had completed the second data lock but showed a severe lack of compliance with the intervention, as assessed by the individual log-in history (i.e. no evidence of engagement with the intervention, as signalled by the absence of any log-ins to the website), were excluded. Notably, since never logging-in cannot possibly be influenced by the intervention assigned to them (because such participants will have never accessed their assigned materials), and is an objective datum, we do not expect this exclusion criterion to introduce any bias in the design. No other post-randomization exclusion criteria were applied for the purpose of the preregistered analysis.

### 2.7.2. Preregistered analysis on the primary outcome

The proportion of participants reporting low-to-severe anxiety (GAD-7 score $\geq 5$) and/or post-traumatic symptomatology (IES-R score $\geq 9$) at the one-week follow-up was compared between the two groups using a unidirectional $\chi^2$-test. If $p < 0.05$ our conclusion would be that SH+ is more effective than the alternative activity for lowering the proportion of care home workers reporting anxiety and/or post-

traumatic symptomatology by the end of the intervention period. If $p \geq 0.05$, our conclusion would be that there is no evidence that SH+ is more effective than the alternative activity for lowering the proportion of care home workers reporting anxiety and/or post-traumatic symptomatology by the end of the intervention period.

### 2.7.3. Additional analyses on the primary and secondary outcomes

#### 2.7.3.1. Additional analyses on the primary outcomes

The percentage of participants completing both scales and passing the threshold for mild symptoms at IES-R ($\geq$9) and/or GAD-7 ($\geq$5) at the 14-week follow-up was also compared through a unidirectional $\chi^2$-test. We then performed the Cochran's omnibus Q-test for matched observations [56], to check for the presence of significant differences in percentages across time, both globally and within each group (in case of global significance, by also performing *post hoc* tests on each pair).[7]

The two primary outcome scales were considered separately in sensitivity analyses, both at post-intervention and at the 14-week follow-up. In case data were available for only one of the scales contributing to the primary outcome, such participants would be included in the secondary analysis concerning their available data.

First of all, $\chi^2$-tests were performed for the effect of SH+ on GAD-7 and IES-R separately, by using the same cut-off threshold used for the main outcome. In order to take multiplicity into account, we used the Hochberg correction: since two separate tests were implemented, this implied that the effect of the SH+ on one scale would be considered statistically significant, regardless of its effect on the other scale, if reaching the 0.025 *p*-value threshold, but that the SH+ would be considered as significantly better than the alternative intervention for both scales,[8] in the case of both tests reaching the conventional significance threshold. Risk ratios, with their CIs, were also calculated for either outcome scales in both time periods. This was also done for the primary outcome.

Then, both at post-treatment and at the 14-week follow-up, we tested the effect of SH+ on GAD-7 and IES-R by performing seemingly unrelated regression (SUR) equations [57], controlling for their baseline values. In the case of joint statistical significance of the coefficients related to treatment status, the effect of treatment on each score would be evaluated separately.

#### 2.7.3.2. Additional analyses on the secondary outcomes

In case data were available for only one or two of the secondary outcome scales, participants would be included in the analyses concerning their available data. The effect of SH+ on WHO-5, CD-RISC and PSS was tested by performing SUR equations both at post-treatment and at the 14-week follow-up, controlling for their baseline values. In the case of joint statistical significance of the coefficients related to treatment status, the effect of treatment on each score would be evaluated separately. For both time points, the percentage of participants dropping out from the test was compared between treatment arms through a bidirectional $\chi^2$-test, and the risk ratios with their CIs were calculated.

Perceived effectiveness, assistant helpfulness, commitment,[6] engagement and fulfilment of expectations were compared across the two treatment arms. This was achieved by performing SUR equations and, in case of significance at the 0.05 level, proceeding to simple regressions on treatment status, using two-sided *t*-tests. In all SUR equation models, in case of observations with only one of the two scales available, the Stata command 'suregub' was used to allow for unbalanced data [58].[9] In order to access the primary outcome scales and the secondary outcome scales, participants had to complete the demographic questionnaire, including questions about expectations, commitment, assistant helpfulness and perceived effectiveness. Therefore, such data were available for all participants included in our analyses, except for a few who chose not to give a rating.[10]

---

[7]This was added after seeing the data, to gain insight into the temporal trajectory of the outcome in either group.

[8]The previous description of the Hochberg correction for the situation where multiple treatments are independently compared to a placebo (which clearly does not apply to our design) has now been replaced with the correct description of the test (i.e. use of the Hochberg correction to assess the statistical significance of the SH+ intervention on two distinct outcome measures), as planned before data collection.

[9]This is a clarification. A global *p*-value of 0.05 was used for the SUR equation model and two-sided tests; the Stata command enabled us to perform this model with an unbalanced panel, in case the number of observations differed between regressions.

[10]The previous wording would mistakenly imply there would be no missing values in non-clinical outcomes. Based on the expectation that not all of the randomized participants would access or complete the intervention, an extra response option (I cannot answer), was made available (see Materials available on the OSF).

### 2.7.3.3. Analyses on imbalance at baseline

Descriptive statistics of baseline values of clinical scales, sociodemographic factors and variables related to participants' expectations were performed. Baseline values of sociodemographic factors and variables related to participants' expectations were compared across the two groups. The value of the standardized mean difference (SMD) of 0.1 (in absolute value) was used as the threshold for lack of balance.

Number of log-ins was dicothomized (only one log-in versus multiple log-ins), as well as the variable related to coming directly into contact with COVID-19-positive colleagues or patients in the last two weeks (Yes versus No), reading habits for novels and short stories (at least three versus less than three per year) and appreciation for novels and short stories (finding them 'quite a bit' or 'extremely' versus at most 'moderately' useful). Age at enrolment was grouped into four categories (18–30, 31–40, 41–50, 51+), while level of education (at most Middle school, High School, University) and job title (administrative, healthcare, technical) were grouped into three. The other variables considered in the balance analysis were sex, expected effectiveness and the clinical outcomes (in the case of GAD and IES-R, the binary variables were considered, as well as their combination, corresponding to the baseline value of the main outcome).[11]

Any variables showing imbalance were included in the secondary regression analyses (in the case of binary outcomes, using Poisson regression with robust standard errors [59]). In all regressions with interval-level outcomes, White's test for homoskedasticity against unrestricted forms of heteroskedasticity was used to assess whether including robust standard errors [60].[12]

## 3. Results

### 3.1. Data pre-processing, randomization and RCT sample size

Twenty-two NCH, recruited via convenience and snowball sampling, participated in the study. Across NCH, a total of 443 individual responses were collected. After screening for incomplete surveys ($N = 81$; of which 73 interrupted soon after having provided consent, eight after having completed the GAD-7 questionnaire), surveys in which two or more of the inattentive/careless response check items were not responded to correctly ($N = 0$), surveys from NCH outside of Northern Italy ($N = 0$), a total of 362 complete surveys were retained. Before proceeding to randomization a confirmation request of the intention to participate in the intervention was sent via email/WA/text to respondents. This aimed to check the validity of the contact details provided and to alert respondents that the five-week intervention would start soon. Participants who did not respond or withdrew at this stage ($N = 116$) were not included in the randomization. Individuals who confirmed but had indicated a job type that did not belong to any of the categories of interest were still given access to the intervention but were not included in the randomization ($N = 5$). A respondent ($N = 1$) whose job type did belong to a category of interest but who was part of our research team, hence was not blind to the experimental manipulation, was also excluded from randomization. Finally, only one (the first to enrol[13]) of any two or more responding workers from NCH who shared their accommodation with one another were entered into the randomization. The others were given access to the same intervention that had been randomly assigned to the participant, to prevent contamination between interventions. This led to exclusion of a further two ($N = 2$) respondents from randomization (figure 2).

A total of 238 individuals were included in the randomization (119 were assigned to SH+, 119 to the alternative intervention, thus exceeding our initial recruitment target of 105 per group), which was stratified by recruitment centre. On average, there were 11 (s.d. = 5; range: 4–20) participants per recruitment centre. The randomization script is available on the OSF [31].

As detailed in figure 2, participants with the main outcome available at the one-week follow-up were 170, with a perfect balance between the two groups. In each group, 85 individuals completed the one-week follow-up, which is just below our initial target of 89 per group. Participants with the main outcome available at the 14-week follow-up were 143 (68 for SH+ and 75 for the alternative intervention).

The vast majority of randomized participants presented at least mild symptoms at baseline (87.39%), were female (88.24%) and healthcare workers (75.63%); most of them had read three or more novels or

---

[11]The complete list of variables was not specified before data collection. We partly opted for categories consistent with our previous work [5] and partly created categories that would contain at least 10 observations each. In order to create the latter, we needed to see the data first.

[12]This is a clarification. We have now specified how we would control for covariates in regressions with binary outcomes, and whether we would use robust standard errors or not for interval outcomes. We have now added this information in the Methods section.

[13]This specification is needed for completeness and replicability.

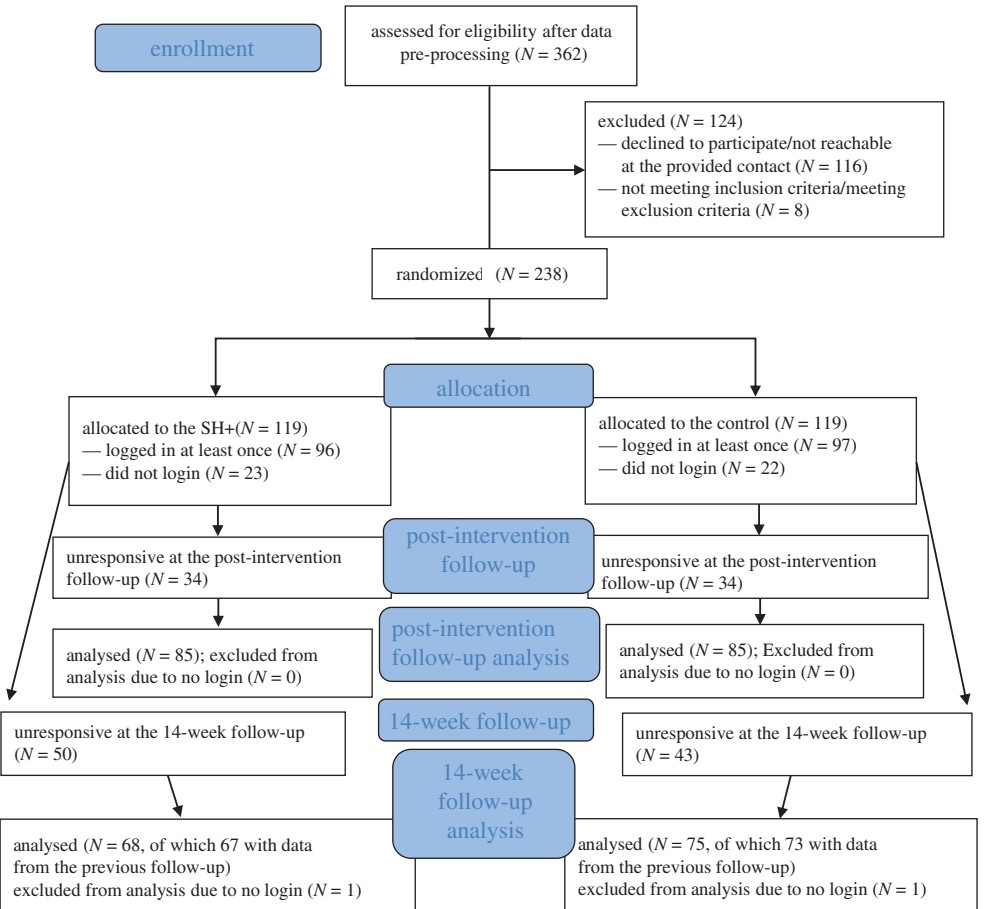

**Figure 2.** CONSORT flow diagram showing the number of participants included in the randomization and analyses, and the number of dropouts per intervention at follow-ups.

short stories in the year before baseline assessment (53.78%) and had found reading novels or short stories useful (64.71%); the most represented age-group was 41–50 (36.13%) and the most frequent highest educational title was the High School degree (46.64%). Twenty-nine per cent (28.99%) of workers who were randomized had had contacts with people affected by COVID-19 in the two weeks preceding the first data log. Among those workers who logged in to the website, 79.79% did it more than once. Concerning the clinical scales, mean score values for the randomized sample were: 6.86 (s.d. = 4.07) for GAD, 23.96 (s.d. = 14.86) for IESR, 13.57 (s.d. = 5.66) for WHO-5, 16.49 (s.d. = 6.70) for PSS and 60.73 (s.d. = 16.59) for CDRISC.

Analyses of baseline values showed imbalance between the group receiving the SH+ and the group receiving the alternative intervention for the percentage of people with at least mild conditions, as measured both by GAD-7 (standardized difference—s.d. −0.14) and by the composite outcome (s.d. −0.11), with more people over the threshold in the alternative intervention group; aged 31–40 (s.d. 0.18), with more people aged 31–40 in the SH+, and those having a High School (s.d. −0.15) or a University (s.d. 0.13) degree, with more people having a High School degree in the alternative intervention group and more people having a University degree in the SH+ group. The balance of all key demographic and clinical characteristics between the two groups is detailed in table 3.

## 3.2. Preregistered analysis on the primary outcome

As shown in table 4, at the post-intervention follow-up, participants assigned to the SH+ showed at least mild symptoms in 63 cases out of 85 (74.12%), whereas participants assigned to the alternative intervention showed at least mild symptoms in 65 cases out of 85 (76.47%). The difference was not significant ($\chi^2 = 0.127$, z-value −0.356, p-value 0.361).

**Table 3.** Analysis of imbalance for clinical, demographic and additional variables between the two groups at baseline. The table includes all randomized participants (i.e. including those with no recorded logins to the intervention website). Bold values in the SMD column indicate imbalance.

| analysis of imbalance | SH+ | alternative | SMD |
|---|---|---|---|
| *clinical variables* | | | |
| main | | | |
| at least mild symptoms (*n/N*) | (101/119) | (107/119) | **−0.11** |
| % | 84.87% | 89.92% | |
| secondary | | | |
| GAD at baseline: at least mild symptoms (*n/N*) | (78/119) | (89/119) | **−0.14** |
| % | 65.55% | 74.79% | |
| IESR at baseline: at least mild symptoms (*n/N*) | (96/119) | (102/119) | −0.09 |
| % | 80.67% | 85.71% | |
| WHO-5 at baseline: mean (s.d.) | 13.58 (5.69) | 13.56 (5.66) | 0.00 |
| PSS at baseline: mean (s.d.) | 16.28 (6.98) | 16.70 (6.44) | −0.04 |
| CDRISC at baseline: mean (s.d.) | 61.88 (17.56) | 59.59 (15.56) | 0.10 |
| *other variables* | | | |
| gender female (*n/N*) | (104/119) | (106/119) | −0.04 |
| % | 87.39% | 89.08% | |
| age 18–30 (*n/N*) | (15/119) | (17/119) | −0.03 |
| % | 12.61% | 14.29% | |
| age 31–40 (*n/N*) | (31/119) | (19/119) | **0.18** |
| % | 26.05% | 15.97% | |
| age 41–50 (*n/N*) | (39/119) | (47/119) | −0.10 |
| % | 32.77% | 39.50% | |
| age 51 or more (*n/N*) | (34/119) | (36/119) | −0.03 |
| % | 28.57% | 30.25% | |
| at most Middle School (*n/N*) | (25/119) | (22/119) | 0.04 |
| % | 21.01% | 18.49% | |
| High School (*n/N*) | (49/119) | (62/119) | **−0.15** |
| % | 41.18% | 52.10% | |
| University (*n/N*) | (45/119) | (35/119) | **0.13** |
| % | 37.82% | 29.41% | |
| administrative staff (*n/N*) | (16/119) | (11/119) | 0.09 |
| % | 13.45% | 9.24% | |
| healthcare (*n/N*) | (89/119) | (91/119) | −0.03 |
| % | 74.79% | 76.47% | |
| technical staff (*n/N*) | (14/119) | (17/119) | −0.05 |
| % | 11.76% | 14.29% | |
| COVID-19 contacts (*n/N*) | (31/119) | (38/119) | −0.09 |
| % | 26.05% | 31.93% | |
| three or more stories (*n/N*) | (65/119) | (63/119) | 0.02 |
| % | 54.62% | 52.94% | |

(*Continued.*)

**Table 3.** (*Continued.*)

| analysis of imbalance | SH+ | alternative | SMD |
|---|---|---|---|
| stories useful (*n/N*) | (79/119) | (75/119) | 0.05 |
| % | 66.39% | 63.03% | |
| effectiveness expected: mean (s.d.) | 6.69 (2.06) | 6.92 (1.83) | −0.08 |
| login once only[a] (*n/N*) | (21/96) | (18/97) | 0.06 |
| % | 21.88% | 18.56% | |
| observations | **119** | **119** | — |

[a]Among those who have at least one login.

**Table 4.** Descriptive statistics and test results for our main binary outcomes. The primary outcome is shown in bold in the first row. A positive *z*-value indicates a higher percentage in the SH+ group, a negative *z*-value indicates a higher percentage in the alternative group.

| analysis without controlling for imbalance at baseline | SH+ | alternative | *z*-value (*p*-value) | RR (CI) |
|---|---|---|---|---|
| *post-intervention* | | | | |
| **frequency of mild symptoms in at least 1 scale (N = 170)** | **63/85 (74.12%)** | **65/85 (76.47%)** | **−0.356 (0.361)** | **0.969 (0.816–1.151)** |
| frequency of mild to severe gad (N = 170) | 49/85 (57.65%) | 48/85 (56.47%) | 0.155 (0.562) | 1.021 (0.786–1.325) |
| frequency of mild to severe IES-R (N = 170) | 59/85 (69.41%) | 62/85 (72.94%) | −0.508 (0.306) | 0.952 (0.786–1.152) |
| dropouts at (N = 238) | 34/119 (28.57%) | 34/119 (28.57%) | 0.000 (1.000) | 1.000 (0.669–1.494) |
| *14-week follow-up* | | | | |
| frequency of mild symptoms in at least 1 scale (N = 143) | 44/68 (64.71%) | 56/75 (74.67%) | −1.297 (0.097) | 0.867 (0.696–1.079) |
| frequency of mild to severe GAD (N = 144) | 33/69 (47.83%) | 41/75 (54.67%) | −0.820 (0.206) | 0.875 (0.634–1.206) |
| frequency of mild to severe IES-R (N = 143) | 37/68 (54.41%) | 46/75 (61.33%) | −0.838 (0.201) | 0.887 (0.669–1.176) |
| dropouts at (N = 238) | 51/119 (42.86%) | 44/119 (36.97%) | 0.926 (0.354) | 1.159(0.847–1.585) |

## 3.3. Additional analyses on the primary and secondary outcomes

### 3.3.1. Primary outcomes

At the second follow-up, the percentage of participants passing the threshold for mild symptoms at the IES-R and/or the GAD-7 was 64.71% (44/68) for the SH+ and 74.67% (56/75) for the alternative intervention ($\chi^2 = 1.683$, *z*-value −1.297, *p*-value 0.097).

Figure 3 shows the trend of the main outcome over time for the two treatment groups separately. The Cochran's test (see footnote 6) was statistically significant (*p*-value < 0.001 in all cases) both for each group separately and globally. Statistical significance (*p*-value 0.001 or lower) was also confirmed in each pairwise comparison, apart from the test between T1 and T2 in the alternative group (*p*-value 0.072).

By considering each component of the main outcome separately, no significant difference was found between groups either. In particular, at the post-intervention follow-up , 49/85 individuals who received

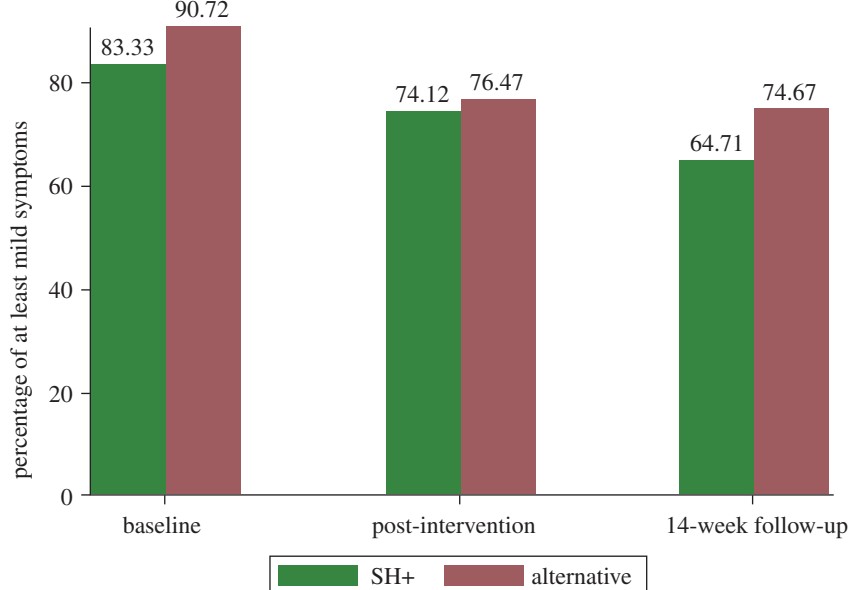

**Figure 3.** The figure shows the percentage of at least mild symptoms of anxiety and/or post-traumatic stress, separately for the two arms of the RCT, across time (baseline, post-intervention follow-up and 14-week follow-up). The figure shows data from randomized participants who logged in at least once.

SH+ versus 48/85 who received the alternative intervention (57.65% versus 56.47%) reported at least mild symptoms in the GAD-7 ($\chi^2 = 0.024$, z-value 0.155, p-value 0.562), while 59/85 versus 62/85 (69.41% versus 72.94%) reported at least mild symptoms at the IES-R ($\chi^2 = 0.258$, z-value −0.508, p-value 0.306).

At the 14-week follow-up, at least mild symptoms were reported in the GAD-7 by 47.83% of the participants (33/69) who received SH+ and by 54.67% of the participants (41/75) who received the alternative intervention ($\chi^2 = 0.673$, z-value -0.820, p-value 0.206), while at least mild symptoms were reported in the IES-R by 54.41% of participants (37/68) who received SH+ versus 61.33% (46/75) who received alternative intervention ($\chi^2 = 0.702$, z-value −0.838, p-value 0.201). Results on the primary outcome and on its components at the two follow-ups are shown in table 4.

Joint analyses on the GAD-7 and IES-R scores controlling for baseline values did not show any statistical significance (p-value 0.544 at T1 and 0.254 at T2) in the SUR equation model (see electronic supplementary material).

No significant results were found even when controlling for imbalanced covariates (see electronic supplementary material).

### 3.3.2. Secondary clinical and additional outcomes

Global tests on secondary outcomes (WHO5, PSS and CD-RISC) did not meet significance (p-value 0.702 at the post-intervention and 0.812 at the 14-week follow-up). The rate of participants dropping out from the study was identical for the two interventions at the post-intervention follow-up (i.e. 28.57%, p-value = 1) whereas, at the 14-week follow-up, SH+ showed a dropout rate of 42.86% (51/119) and the alternative intervention of 36.97% (44/119; see figure 2).[14] Such difference was not statistically significant ($\chi^2 = 0.858$, z-value −0.926, p-value 0.354).

Perceived intervention effectiveness, assistant helpfulness, commitment, engagement and expectation fulfilment turned out as globally significant both at the post-intervention (p-value 0.008) and at the 14-week follow-up (p-value 0.024). Separate tests highlighted significant results at the post-intervention follow-up for perceived effectiveness (ß 0.939, p-value 0.020), expectation fulfilment (ß 1.136, p-value

[14]In order to check the robustness of our results for a possible bias due to informative dropouts, we have included in our electronic supplementary materials the results of a linear mixed model (LMM), that simultaneously estimates the treatment effect on the outcome at all timepoints (T1 and T2, in our case) and produces unbiased treatment effect estimates in case of missingness at random (MAR) [61].

**Table 5.** Descriptive statistics and simple regression results for additional continuous outcomes.

| outcome | SH+: mean (s.d.) | alternative: mean (s.d.) | ß coefficient (CI) | standardized ß coefficient (s.e.) |
|---|---|---|---|---|
| *post-intervention* | | | | |
| perceived effectiveness (N = 155) | 6.22 (2.45) | 5.28 (2.54) | 0.939 (0.147; 1.731)* | 0.186 (0.079) |
| assistant helpfulness (N = 158) | 6.51 (2.67) | 6.04 (2.61) | 0.467 (−0.364; 1.299) | 0.089 (0.080) |
| commitment (N = 155) | 5.87 (2.33) | 5.83 (2.53) | 0.044 (−0.727; 0.816) | 0.009 (0.081) |
| engagement (N = 157) | 6.53 (2.55) | 5.41 (2.58) | 1.123 (0.314; 1.932)* | 0.215 (0.078) |
| fulfilment of expectations (N = 153) | 5.92 (2.48) | 4.79 (2.65) | 1.136 (0.317; 1.955)** | 0.218 (0.079) |
| *14-week follow-up* | | | | |
| perceived effectiveness (N = 136) | 6.08 (2.48) | 5.14 (2.60) | 0.933 (0.070; 1.796)* | 0.182 (0.085) |
| assistant helpfulness (N = 133) | 5.73 (2.80) | 4.94 (2.81) | 0.787 (−0.176; 1.751) | 0.140 (0.087) |
| commitment (N = 134) | 6.17 (2.31) | 6.43 (2.35) | −0.257 (−1.054; 0.540) | −0.055 (0.087) |
| engagement (N = 135) | 6.02 (2.54) | 5.34 (2.48) | 0.673 (−0.182; 1.527) | 0.134 (0.086) |
| fulfilment of expectations (N = 135) | 5.88 (2.41) | 5.33 (2.58) | 0.548 (−0.303; 1.400) | 0.110 (0.086) |

*$p < 0.05$; **$p < 0.01$.

0.007) and engagement (ß 1.123, *p*-value 0.007) and at the 14-week follow-up for perceived effectiveness (ß 0.933, *p*-value 0.034). All these differences were in favour of the SH+ (table 5).

Controlling for imbalanced covariates showed similar results, except for at the 14-week follow-up, where no variable was found to be statistically significant (see electronic supplementary material).

## 4. Discussion

To the best of the authors' knowledge, this is the first randomized study that examines the effects of the SH+ intervention, in its version as *Doing What Matters in Times of Stress*, on decreasing symptoms of anxiety and post-traumatic stress. It is the first to examine the effects of the SH+ in healthcare staff, to adopt a supervised online delivery mode and also to adopt, as a means of comparison, an alternative intervention with similar requirements to the SH+ in terms of time and commitment. The primary outcome consists of the rate of self-reported anxiety and/or post-traumatic stress symptoms at the end of the intervention period. In the preregistered analysis, no significant difference is found in the rate of self-reported symptoms of anxiety and/or post-traumatic stress between the group receiving the SH+ and the group receiving the alternative intervention. In exploratory analyses, perceived intervention effectiveness, match with expectations and engagement are higher at the post-intervention assessment after receiving the SH+ than after receiving the alternative intervention. No difference is found between groups in terms of dropout rate.

As far as the negative result of our preregistered analysis is concerned, we can exclude the presence of insufficient statistical power to detect an effect of applicative interest, which is what we had tuned our recruitment efforts on,[15] given that the rate of mild-to-severe symptoms in the alternative activity

---

[15]A very small, non-significant advantage was found for the SH+ intervention: if the rates of self-reported symptoms found after the intervention were the ones in the population, 11 508 observations would be needed to achieve a 90% power with the one-sided test. On the other hand, the percentages observed at the 14-week follow-up and the results of the unregistered Cochran's Q-test suggest that future studies, adequately powered to identify a possible effect at 14 weeks, would be needed in order to draw any conclusions.

group and the number of participants returning to the post-intervention follow-up were fairly close to our initial target. Several reasons may explain the lack of a difference between arms in our study. A plausible explanation may be due to the chosen type of intervention with which to compare the SH+. Indeed, even if the alternative activity was not expected to exert a direct impact on the ability to manage stressful situations, it consisted of a set of visual and oral materials and exercises, which were accessed by the same dedicated website, and required a similar amount of time and commitment as the SH+ intervention. This design has the advantage of enabling firm conclusions on the possible effects of the intervention based on its specific ACT-based contents [14,15], rather than on non-specific components like receiving attention, or being involved in generic activities [62]. The present findings would suggest that the positive effects of the SH+ observed in previous trials [18,63,64], that did not employ a control condition with similar quality and quantity of interactions and stimulation given to participants, might be related to non-specific factors. These factors were similar to the so-called 'common factors' of psychotherapeutic interventions, such as the provision of structured and interactive activities, contents that stimulate reflection and expression of emotions, guided exercises and the stimulation of multiple senses through visual and auditory channels [65] (see §2.2).

There may be several additional reasons for the lack of a difference between the SH+ and the alternative activity in terms of self-reported symptoms in the current trial. Among them are: the short duration of the intervention and its focus on coping strategies, promotion of flexibility and practical exercises to deal with stress instead of symptoms. Indeed, such characteristics could have reinforced a general sense of intervention effectiveness, and provided general means that could become useful in the long term, without working specifically on psychological symptoms, as more structured and more extensive psychotherapies may do [66]. Moreover, our results might also be related to the characteristics of the target population. As mentioned in the Introduction, a RCT testing the effectiveness of the standard SH+ intervention versus usual care (i.e. no alternative activity) on 694 refugee women in Uganda found significant results in lowering mental health symptoms at post-intervention, and at the three-month follow-up [18]. Similarly, an RCT involving 642 asylum seekers and refugees resettled in a refugee camp in Turkey identified a positive effect of SH+ in preventing any mental disorders at a six-month follow-up [64]. However, the Ugandan trial was focused on severely distressed women, and in both Uganda and Turkey the target population was displaced and living in a refugee camp, so that the intervention effect might be linked to the severity of conditions and to the scarcity of alternative interventions or supportive strategies in those contexts [18,64]. Our target population was composed of workers from NCH with higher educational levels and with lower levels of symptoms at baseline, so that the effect of a low-intensity intervention might be less emphasized. On the other hand, our results appear to be relatively more aligned with those of a recent RCT testing the standard SH+ intervention in Europe with a preventative aim [63]. In this trial, 459 asylum seekers and refugees, resettled in five European countries, with increased psychological distress but without a formal psychiatric diagnosis, were randomized to receive the SH+ or enhanced usual care (i.e. no alternative activity). This trial also failed to identify a significant difference for the primary outcome, that was the frequency of any mental disorders at six-month follow-up. However, the authors did report, as a secondary result, a positive effect for psychological distress, depressive symptoms, perceived problems and well-being at post-intervention. For well-being (as measured with the WHO-5, like in our study), the estimate was still significant at the six-month follow-up [63].

In exploratory analyses, there was a potentially interesting finding, that is, significant differences between the two intervention arms in perceived intervention effectiveness, expectation fulfilment and engagement at the post-intervention assessment, on a background of identical drop-out rates. All the differences favoured the SH+. As the SH+ explicitly provided tools and strategies to cope with adversities that were not part of the alternative intervention, this feature may have enhanced perceived effectiveness by increasing a sense of self-efficacy. Self-efficacy relates to a person's perceived ability to perform well in challenging situations and to manage environmental demands [67]. A greater self-efficacy perception to cope with stressors has a positive influence on human functioning and might help, to some extent, to protect against stress and anxiety and to promote mental health in the long term [67,68]. Perceived effectiveness has been linked to patient satisfaction in a healthcare context, health improvement and maintenance of a healthy status [69], whereas meeting expectations is a critical component of patient satisfaction and it is more likely to occur when a treatment is considered effective [70]. Both the perception of effectiveness and fulfilment of expectations might thus be associated with long-term improvements in mental health. Since the SH+ was found to be superior for these characteristics (albeit in secondary analyses), even in a conservative approach like ours where the SH+ group was compared to an active control group, it may be worth

testing whether it could be useful for skilled healthcare staff working in long-term healthcare facilities beyond emergency situations.

To conclude, our study has several strengths. These include the random allocation, the choice of an active control group, the fact that the participants and the majority of the researchers involved were blind to the intervention allocation, and that the website manager was blind to participants' baseline data. Moreover, we provided a large sample of workers in NCH with a practical strategy to independently manage distress, which was perceived—according to secondary analyses—as more effective and engaging than an alternative but similarly demanding activity. It could be interesting to investigate these differences in perception, which here represented only secondary outcomes, in a study that focuses on them and their expected benefits as the main outcome. Indeed, exposure to chronic stressors and multiple traumatic experiences may generate long-term consequences even after the acute phase of the pandemic, especially for workers in NCH [71,72]. Our study also has several limitations. These include reliance on self-reported instruments, that may impact the evaluation of psychological symptoms. Further, despite the fact that the drop-out rates were very similar between groups, proving intervention feasibility, we were not able to identify and analyse reasons related to the dropouts. Another limitation is the lack of longer-term assessments, which could have provided important insights for potential implementation, including, for example, the use of booster sessions to reinforce the intervention effect. This is particularly important, in consideration of the variable results present in the literature on the SH+, in terms of the timing of its potential effectiveness. Given the negative findings of this study for the primary outcome, before the SH+ in its *Doing What Matters in Times of Stress* is implemented in clinical practice, further studies should be conducted to examine both its short- and long-term beneficial effects, by means of randomized studies that employ alternative but similarly structured interventions as control conditions, aiming to minimize the confounding effect of non-specific factors.

Ethics. The clinical research Internal Review Board of Gruppo Servizi Pastorali Educativi Sociali (SPES) approved the protocol. The trial was conducted according to globally accepted standards of good clinical practice (as defined in the ICH E6 Guideline for Good Clinical Practice, 1 May 1996), in agreement with the Declaration of Helsinki. Informed consent was obtained from all subjects at study enrolment. All participants could interrupt their participation and/or withdraw from the study at any time.

Data accessibility. The article's supporting data and digital research materials (with the exception of the CD-RISC 25, which is protected by copyright but can be obtained from the authors upon signing a user agreement: http://www.connordavidson-resiliencescale.com/index.php) are available through this dedicated link [31]: https://osf.io/gdpsw/. Following in-principle acceptance, the approved Stage 1 version of this manuscript was preregistered on the OSF at https://osf.io/ep5mk. This preregistration was performed prior to data collection and analysis.

Competing interests. We have no competing interests.

Funding. We received no funding for this study.

Acknowledgments. The authors would like to thank Luca Lorenzi who kindly lent his voice to record the audio files for the alternative activity.

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
