## [Peer Review File · Royal Society Open Science]

Review History

RSOS-210219.R0 (Original submission)

Review form: Reviewer 1

Do you have any ethical concerns with this paper?

No

Recommendation?

Accept with minor revision

Comments to the Author(s)

This is an interesting piece of research making use of the COVID-19 pandemic to evaluate an important research question with clinical implications. The hypotheses were clear, and the methodology is plausible to allow the hypothesis to be answered. The quality checks were

sufficient. The methodology of the study was detailed and would allow further replication. A few minor comments are detailed below:

- (1) The introduction could be more concise, there was a lot of detail that was later repeated in the methods section.
- (2) When discussing the ACT components of the intervention in the introduction it reads as if the authors are planning to test the active components of the intervention or test it against another active intervention (e.g. "The potential beneficial effects of SH+ on psychological symptoms may be related to its main core components [...] Accordingly, ACT might lead to broader substantial changes than other forms of psychological interventions regarding psychological functioning"). It would be helpful to rephrase this if this is not the author's intention.
- (3) Further justification of the anticipated effect would be helpful. Has a reduction of 20% been based on previous literature?
- (4) The exact sample size the authors intend to recruit is currently unclear. The authors report a sample size calculation of 82 participants per group after attrition, but are unclear what the expected attrition rate is. It would be best to estimate an attrition rate based on similar trials and account for this in recruitment to ensure the study is not underpowered.
- (5) The authors may consider registering some of the additional outcomes currently listed as exploratory as secondary analyses to increase confidence in their findings.

Review form: Reviewer 2

Do you have any ethical concerns with this paper?

No

Recommendation?

Major revision

Comments to the Author(s)

Attached are my comments (see Appendix A) and a CONSORT checklist completed to the end of the methods section (see Appendix B).

Review form: Reviewer 3

Do you have any ethical concerns with this paper?

No

Recommendation?

Accept with minor revision

Comments to the Author(s)

The manuscript "Effectiveness of Self-Help Plus (SH+) in reducing anxiety and posttraumatic symptomatology among care home workers during the COVID-19 pandemic" (RSOS-210219) proposes the evaluation of a WHO-developed intervention package. The manuscript generally reads well and the major points to pin-point the main analysis have been described. The evaluation of the efficacy of a programme in a new target population is a valid research question. There are some aspects I comment on in more detail below, mainly for documentation of the evaluated aspects, but I want to highlight that the key questions I have are (i) why an surprisingly

simple trial analysis was chosen, (ii) why contamination at centre-/workplace-level is accepted, and (iii) why the primary outcome is evaluated at such a short time frame (1-week).

1) Primary outcome

1a) The power calculation was performed in proprietary software and could not be replicated. Based on the OSF-provided code the calculation is correct. The complete syntax for the planned test, also in line with the description in chapter 2.7.2 (making 5% sig level and the planned chi2 test explicit), would have been:

```
power twoproportions 0.8 0.6, power(0.8) alpha(0.05) test(chi2)
```

Using other calculators with these specifications results in the described target sample size.

1b) The team should mention that the test is two-sided (often forgotten when dealing with chi2 distributions).

1c) The section in 2.7.2 should entail a statement about how the result of the test will be interpreted. As the test is two sided, it is not pre-defined which constellation of proportions and significance test result will be interpreted as in line with the hypothesised effect of the intervention.

1d) While the logic of the testing strategy is sound, it rests on a number of assumptions and decisions that have already been made. The dichotomisation of the outcome is usually considered a severe limitation in the trials literature and should be avoided where possible (as it also decreases power). The team is also not controlling for anxiety and posttraumatic stress symptoms at the start of the study, which is counter-intuitive given the detailed introduction provided around the development of stress levels in the target population. This latter point is again important as it again reduces statistical power and it will make the interpretation of the results more difficult, as groups can obviously start at different proportions even with randomisation present, i.e. the endpoint alone is not on its own relevant for the decision on the effectiveness of the programme (important for 1c as well). I want to stress: within the logic of the study this is fine, but a stronger testing strategy could have been developed.

1e) With 1-week, the evaluation of the effectiveness is on the extreme short-term side for psychosocial workplace interventions, in particular self-guided ones. More justification for this choice could have been provided.

1f) Using a composite measure (and a dichotomous one, see 1d) is usually fraught with difficulties. But due to its logical definition as "either type of symptom" the typical problems of one dimension changing but not the other etc are avoided.

2) Experimental design

2a) If I understood 2.1.4 correctly, the study does not address contamination sufficiently. While contamination at household level is avoided (2.1.4, line 2), contamination at centre level (healthcare workers talking to each other, sharing materials etc) is not addressed at all. This seems to me to be a severe shortcoming of the design.

2b) I may have missed it, but I did not find a description of how the data will exactly be collected (which online system, who has access at which point etc).

2c) I don't think it is stated how many potential participants will be approached or whether this study will approach as many as needed to get to the required number of responses (as per power analysis).

3) Intervention and control

I checked the report of the intervention and control conditions against the TIDieR checklist. The only thing missing were the materials for the control group (the link provided led only to intervention group materials).

MINOR:

I am not sure what the team means with the "two data locks". Do the authors mean that the data were logged at those time points?

To leverage the power of the pre-registration, the planned analysis for the secondary outcomes could be described as well. It sounds like the authors have already settled on a plan for action, i.e. it could easily be added.

page 13: "A facilitator will check in regularly with the participants via WhatsApp, text messages and phone calls, to monitor their progress and to encourage compliance" Will this be done in both groups?

Decision letter (RSOS-210219.R0)

Dear Professor Rusconi

On behalf of the Editors, I am pleased to inform you that your Manuscript RSOS-210219 entitled "EFFECTIVENESS OF SELF-HELP PLUS (SH+) IN REDUCING ANXIETY AND POST-TRAUMATIC SYMPTOMATOLOGY AMONG CARE HOME WORKERS DURING THE COVID-19 PANDEMIC" deemed suitable for in-principle acceptance in Royal Society Open Science subject to minor revision in accordance with the referee and editor suggestions. Please find their comments at the end of this email.

The reviewers and handling editors have recommended publication, but also suggest some minor revisions to your manuscript. Therefore, I invite you to respond to the comments and revise your manuscript.

Please you submit the revised version of your manuscript within 7 days (i.e. by the 25-Feb-2021). If you do not think you will be able to meet this date please let me know immediately.

Full author guidelines can be found here <https://royalsocietypublishing.org/rsos/registered-reports#ReviewerGuideRegRep>.

on behalf of Professor Chris Chambers
(Subject Editor, Royal Society Open Science)
openscience@royalsociety.org

Associate Editor Comments to Author (Professor Chris Chambers):

Four reviewers have now assessed the manuscript, including two field experts (Revs 1 and 3), one expert in trial methodology (Rev 2) and a specialist statistical editor. The reviews overall are cautiously positive. Key issues to address in revision include clarification and justification of key elements of the methodology and rationale, addressing contamination effects, concerns regarding variable dichotomization (raised by multiple reviewers), interpretation of non-significant results (and potential alternative analyses to consider), sufficiency of positive controls, and adherence to trial reporting standards.

Concerning the point raised by reviewer 2: "Before commencing the trial, the authors should register their study on a clinical trial registry. That they will do this and where and when should be stated in the protocol." Note that as part of this RSOS special initiative, the journal preregisters the Stage 1 protocol at the point of in-principle acceptance, and we will follow up with you in due course about that.

Concerning the prespecification of secondary outcomes and analyses, I am open to these being included in the Stage 1 manuscript (as suggested by several reviewers – and there are good arguments for doing so in the context of a clinical trial); alternatively, provided these analyses are going to be strictly exploratory in nature (and not dominate the conclusions) then the authors are welcome to reserve mention of these analyses until Stage 2. This approach differs somewhat from expectations in conventional trial reporting.

Comments from Specialist Statistical Editor

The authors use what is meant to be an active control, to control for in their words, expectation, facilitator attention, and participant engagement. Outcome neutral tests are needed for each of these claims.

For example, for expectation, they could ask participants to give their pretest scores and predict their posttest scores on the outcome DVs, after reading the materials of the intervention, but before the intervention. They could instead run a separate expectation elicitation experiment, where subjects read the materials of each intervention and give the ratings as above - the difference being one can collect expectations before and separate to the main experiment.

Equivalence of expectations could be tested by a Bayes factor where the obtained effect on the crucial outcome variables is used to inform what expectation might explain that outcome. The

simplest hypothesis could use the obtained effect as the SD of a half-normal on the expected effects.

Using a cut off on the scales to make a binary division can considerably reduce power. Why not use the continuous scales themselves? Maybe clinical interest is in the cut offs, but if there is no strong case for using cut offs in terms of clinical interpretation, from a statistical point of view using continuous variables is better.

For power an effect of interest is plucked from the air. Where does this come from? Why not another effect? And as this is a predicted effect not a smallest effect of interest, it should be used for Bayes factors not power. Or else the authors should justify a smallest effect of interest to calculate power.

Reviewer comments to Author:

Reviewer: 1

Comments to the Author(s)

This is an interesting piece of research making use of the COVID-19 pandemic to evaluate an important research question with clinical implications. The hypotheses were clear, and the methodology is plausible to allow the hypothesis to be answered. The quality checks were sufficient. The methodology of the study was detailed and would allow further replication. A few minor comments are detailed below:

- (1) The introduction could be more concise, there was a lot of detail that was later repeated in the methods section.
- (2) When discussing the ACT components of the intervention in the introduction it reads as if the authors are planning to test the active components of the intervention or test it against another active intervention (e.g. "The potential beneficial effects of SH+ on psychological symptoms may be related to its main core components [...] Accordingly, ACT might lead to broader substantial changes than other forms of psychological interventions regarding psychological functioning"). It would be helpful to rephrase this if this is not the author's intention.
- (3) Further justification of the anticipated effect would be helpful. Has a reduction of 20% been based on previous literature?
- (4) The exact sample size the authors intend to recruit is currently unclear. The authors report a sample size calculation of 82 participants per group after attrition, but are unclear what the expected attrition rate is. It would be best to estimate an attrition rate based on similar trials and account for this in recruitment to ensure the study is not underpowered.
- (5) The authors may consider registering some of the additional outcomes currently listed as exploratory as secondary analyses to increase confidence in their findings.

Reviewer: 2

Comments to the Author(s)

Attached are my comments and a CONSORT checklist completed to the end of the methods section.

Reviewer: 3

Comments to the Author(s)

The manuscript "Effectiveness of Self-Help Plus (SH+) in reducing anxiety and posttraumatic symptomatology among care home workers during the COVID-19 pandemic" (RSOS-210219) proposes the evaluation of a WHO-developed intervention package. The manuscript generally reads well and the major points to pin-point the main analysis have been described. The evaluation of the efficacy of a programme in a new target population is a valid research question.

There are some aspects I comment on in more detail below, mainly for documentation of the evaluated aspects, but I want to highlight that the key questions I have are (i) why an surprisingly simple trial analysis was chosen, (ii) why contamination at centre-/workplace-level is accepted, and (iii) why the primary outcome is evaluated at such a short time frame (1-week).

1) Primary outcome

1a) The power calculation was performed in proprietary software and could not be replicated. Based on the OSF-provided code the calculation is correct. The complete syntax for the planned test, also in line with the description in chapter 2.7.2 (making 5% sig level and the planned chi2 test explicit), would have been:

```
power twoproportions 0.8 0.6, power(0.8) alpha(0.05) test(chi2)
```

Using other calculators with these specifications results in the described target sample size.

1b) The team should mention that the test is two-sided (often forgotten when dealing with chi2 distributions).

1c) The section in 2.7.2 should entail a statement about how the result of the test will be interpreted. As the test is two sided, it is not pre-defined which constellation of proportions and significance test result will be interpreted as in line with the hypothesised effect of the intervention.

1d) While the logic of the testing strategy is sound, it rests on a number of assumptions and decisions that have already been made. The dichotomisation of the outcome is usually considered a severe limitation in the trials literature and should be avoided where possible (as it also decreases power). The team is also not controlling for anxiety and posttraumatic stress symptoms at the start of the study, which is counter-intuitive given the detailed introduction provided around the development of stress levels in the target population. This latter point is again important as it again reduces statistical power and it will make the interpretation of the results more difficult, as groups can obviously start at different proportions even with randomisation present, i.e. the endpoint alone is not on its own relevant for the decision on the effectiveness of the programme (important for 1c as well). I want to stress: within the logic of the study this is fine, but a stronger testing strategy could have been developed.

1e) With 1-week, the evaluation of the effectiveness is on the extreme short-term side for psychosocial workplace interventions, in particular self-guided ones. More justification for this choice could have been provided.

1f) Using a composite measure (and a dichotomous one, see 1d) is usually fraught with difficulties. But due to its logical definition as "either type of symptom" the typical problems of one dimension changing but not the other etc are avoided.

2) Experimental design

2a) If I understood 2.1.4 correctly, the study does not address contamination sufficiently. While contamination at household level is avoided (2.1.4, line 2), contamination at centre level (healthcare workers talking to each other, sharing materials etc) is not addressed at all. This seems to me to be a severe shortcoming of the design.

2b) I may have missed it, but I did not find a description of how the data will exactly be collected (which online system, who has access at which point etc).

2c) I don't think it is stated how many potential participants will be approached or whether this study will approach as many as needed to get to the required number of responses (as per power analysis).

3) Intervention and control

I checked the report of the intervention and control conditions against the TIDieR checklist. The only thing missing were the materials for the control group (the link provided led only to intervention group materials).

MINOR:

I am not sure what the team means with the "two data locks". Do the authors mean that the data were logged at those time points?

To leverage the power of the pre-registration, the planned analysis for the secondary outcomes could be described as well. It sounds like the authors have already settled on a plan for action, i.e. it could easily be added.

page 13: "A facilitator will check in regularly with the participants via WhatsApp, text messages and phone calls, to monitor their progress and to encourage compliance" Will this be done in both groups?

Author's Response to Decision Letter for (RSOS-210219.R0)

See Appendix C.

Decision letter (RSOS-210219.R1)

Dear Professor Rusconi

On behalf of the Editor, I am pleased to inform you that your Manuscript RSOS-210219.R1 entitled "EFFECTIVENESS OF SELF-HELP PLUS (SH+) IN REDUCING ANXIETY AND POST-TRAUMATIC SYMPTOMATOLOGY AMONG CARE HOME WORKERS IN THE COVID-19 PANDEMIC: AN RCT" has been accepted in principle for publication in Royal Society Open Science.

You may now progress to Stage 2 and complete the study as approved.

Please read the following email carefully

Your accepted Stage 1 manuscript has been registered under the requested 4-year private embargo at: <https://osf.io/ep5mk>

This embargo will be released, and the accepted Stage 1 manuscript made public, at the point of Stage 2 submission or manuscript withdrawal.

Following completion of your study, we invite you to resubmit your paper for peer review as a Stage 2 Registered Report. Please note that your manuscript can still be rejected for publication at Stage 2 if the Editors consider any of the following conditions to be met:

- The results were unable to test the authors' proposed hypotheses by failing to meet the approved outcome-neutral criteria.
- The authors altered the Introduction, rationale, or hypotheses, as approved in the Stage 1 submission.
- The authors failed to adhere closely to the registered experimental procedures. Please note that any deviations from the approved experimental procedures must be communicated to the editor immediately for approval, and prior to the completion of data collection. Failure to do so can result in revocation of in-principle acceptance and rejection at Stage 2 (see complete guidelines for further information).
- Any post-hoc (unregistered) analyses were either unjustified, insufficiently caveated, or overly dominant in shaping the authors' conclusions.
- The authors' conclusions were not justified given the data obtained.

We encourage you to read the complete guidelines for authors concerning Stage 2 submissions at <https://royalsocietypublishing.org/rsos/registered-reports#ReviewerGuideRegRep>. Please especially note the requirements for data sharing, reporting the URL of the independently registered protocol, and that withdrawing your manuscript will result in publication of a Withdrawn Registration.

Once again, thank you for submitting your manuscript to Royal Society Open Science and we look forward to receiving your Stage 2 submission. If you have any questions at all, please do not hesitate to get in touch. We look forward to hearing from you shortly with the anticipated submission date for your stage two manuscript.

on behalf of Professor Chris Chambers (Registered Reports Editor, Royal Society Open Science)
openscience@royalsociety.org

Author's Response to Decision Letter for (RSOS-210219.R1)

See Appendix D.

RSOS-210219.R2

Review form: Reviewer 1

Is the manuscript scientifically sound in its present form?

Yes

Are the interpretations and conclusions justified by the results?

Yes

Is the language acceptable?

Yes

Do you have any ethical concerns with this paper?

No

Have you any concerns about statistical analyses in this paper?

No

Recommendation?

Accept with minor revision

Comments to the Author(s)

This paper presents the second stage of a registered report on the effectiveness of a self-help intervention (SH +) versus an active control in reducing anxiety and PTSD symptoms in care home workers during the COVID-19 pandemic. From the pre-registered primary analysis, the authors did not find evidence that the SH + intervention was more effective than the active control. This study makes an important contribution to the literature in testing the effectiveness of the SH + intervention in a randomised design with an active comparison group. Based on their findings the authors speculate that previously observed positive effects of the SH + in studies without control groups may be due to non-specific factors.

The authors have made minor changes to the introduction section. For the most part these changes were warranted in updating the text to reflect the current situation. The stated hypotheses are the same as the approved stage 1 submission, and the data are able to test the proposed hypotheses. The authors adhered to the registered experimental procedures except for a 2-week delay to the planned timepoint for the time 2 follow-up. The authors have been transparent about this change and note that it does not influence their primary pre-registered outcome.

I found the results section to be overly dense which made understanding the researcher's findings difficult. I think it would be beneficial to include some of the results in supplementary materials. For example, the authors could note that controlling for imbalance at baseline did not impact their findings in the main manuscript and include the details of these analyses in supplementary materials. It would also be informative to highlight which analyses were registered versus exploratory in the results section.

The tables and figures were somewhat difficult to read in their present format. It would be helpful to either separate figure 3 into two separate figures depending on the outcome (% of mild symptoms and difference in percentage) or just focus on one outcome rather than have two y axes. Adding an additional row or column to highlight the timepoints in table 4 (as the authors have already done in table 5) would be helpful. Using more descriptive labels for the timepoints in the tables and figures would also help clarify the author's findings (e.g., baseline, post-intervention and 14-week follow-up rather than T0, T1, T2).

I have not previously come across the Cochran's omnibus Q test conducted by the authors and am therefore not able to comment on whether this unregistered exploratory analysis was statistically justified or methodologically sound. However, at present I feel that this test is not informative in the way results are presented. I found it confusing that the authors report significant findings for this analysis except for between T1 and T2 in the alternative group, but do

not provide any further comment on this. The descriptive statistics seem to indicate that the whereas the SH+ group showed a decline from T1 to T2 in mild symptoms in at least 1 scale (74.12% to 64.71%; Table 4) the alternative group showed relatively stable levels (76.47% to 74.67%) which would seem to indicate some benefit of the SH+ in the long-term. It may be that I am misinterpreting these findings, but I believe further commentary would be helpful in helping readers to understand the practical significance of this analysis if it warrants inclusion as an exploratory test. Aside from this I feel that the author's conclusions are justified given the data.

Review form: Reviewer 3

Is the manuscript scientifically sound in its present form?

Yes

Are the interpretations and conclusions justified by the results?

Yes

Is the language acceptable?

Yes

Do you have any ethical concerns with this paper?

No

Have you any concerns about statistical analyses in this paper?

No

Recommendation?

Accept with minor revision

Comments to the Author(s)

I thank the editorial team for inviting me to review the Stage 2 version of this paper and I congratulate the team to finish their study so closely within the confines of their registration.

I can confirm that the Introduction, rationale and stated hypotheses are the same as the approved Stage 1 submission. The report adheres to the registered experimental procedures and minimal deviations from the original approach are transparently described and in my view largely (see comments below) within the affordances of such a practice-based research project.

The presented data are able to test the authors' proposed hypotheses and the exploratory analyses are clearly separated in the report. The authors' conclusions are largely justified given the data and reported results (comments on smaller aspects below).

I provide my comments below separated into Major and Minor points as well as a personal observation relating to the results – the latter clearly separated as it may invite what feels like undue speculation on the results and is not a critique based on comparisons with the IPA manuscript.

__MAJOR__

1) Page 19, " Finally, only one (the first to enrol) of any two or more responding workers from NCH who shared their accommodation with one another were entered into the randomization." This is a difficult point that would require at least further discussion as a limitation. This detail of the procedure was not registered – in the Stage 1 manuscript it was only specified that:

p. 7: "Those meeting the inclusion criteria outlined below, completing an initial survey (demographic information and baseline assessment) and passing the validity check at the baseline assessment (see section 2.4) will then be randomly assigned to one of the two activity groups. Randomization will be stratified by centre and the RCT will be conducted in accordance with the Consolidated Standards of Reporting Trials statement."

And it was later specified, p. 8, that:

"To avoid contamination, only one person per household will be randomised."

As conducted and reported, in study planning this should have been an inclusion criterion i.e. stating that the first to enrol from shared accommodation will be randomised.

As conducted now, this is effectively a randomisation breach as the protocol/Stage 1 paper defined everyone as randomizable.

As far as I understand, the likely effect for this trial will be minimal as the authors report on page 19 "This led to exclusion of a further two (N = 2) respondents from randomization (see Figure 2)." Nevertheless, this should at least be noted.

2) p. 27: The discussion of secondary results starting with "In exploratory analyses,..." seems to go a bit far when compared to the registration as well as to the results that are reported in the manuscript.

2a) "Since the SH+ has proved to be predominant for these characteristics, even in a conservative test like ours (albeit in secondary analyses), we can hypothesise that its use might be effective for skilled healthcare staff working in long-term healthcare facilities beyond emergency situations."

2a-i) It is unclear to me what "predominant" means in this case and how this relates to the strong interpretation offered.

2a-ii) It is unclear to me which test is described as 'conservative' here. From a statistical perspective any analysis this can refer to would seem to be pretty liberal as it was not pre-registered.

2b) "Any further speculation on these findings would be unwarranted,..."

It is unclear to me why this small paragraph was included as it contains such 'further speculation'. Especially, the link to stepped care models seems to be a long shot as the present study was not conducted in such a context.

3) p. 28: "Moreover, we provided a large sample of workers in NCH with a practical strategy to independently manage distress, which was perceived - according to secondary analyses - as more effective and engaging than an alternative but similarly demanding activity. If this was confirmed, the strategy could be adapted for many other groups, which is particularly important given the potential long-term consequences of the COVID-19 pandemic."

This interpretation seems to assign undue relevance to the results of analyses that were not pre-planned as the primary outcome. This should either be dropped from the manuscript or the wording revised to reflect the exploratory nature of the results.

4) p. 28 "Exposure to chronic stressors and multiple traumatic experiences may generate long-term consequences even after the acute phase of the pandemic, especially for workers in NCH." I am not sure why this sentence is placed in the "strengths" section.

If kept, there should be a reference provided - the presented empirical statement does not follow from this study.

5) Table 5 and its description are difficult to match to the description of the methods applied in 2.7.3.2. If I interpreted correctly that these are the results connected to that part of the methods section, the linkage in terminology could be increased (also: the SUR results which are indicated as the first methodological step seem not to be reported).

MINOR

6) What is the RSOS convention for decimal places as a number of different formats are used.

7) p. 22, "The Cochran's test turned out to be statistically significant..."
Suggestion: "was statistically significant"

8) p. 24: "Joint analyses on the GAD-7 and IES-R scores controlling for baseline values did not show any statistical significance (p-value 0.544 at T1 and 0.254 at T2) in the SUR equation model." Partly a question to the editor: Is it admissible to report results without reporting full results (coefficients, SEs, and in case of SUR especially the residual correlations)?

9) p. 28: "...the blindness of participants and of the majority of the researchers involved..."
I suggest to reformulate, as neither participants and researchers were physically blind (I presume...)

10) Table 2 is not introduced/linked in the text.

11) Table 4 contains drop-out ratios, which seem not to be commented on/explained in the text accompanying the table.

12) Page 18, "Number of log-ins was dic[h]othomized (no log-ins vs. at least one log-in),..."
Typo as indicated and it is not clear why the distribution of the raw data is not presented as well.

13) p. 26 "In exploratory analyses, perceived intervention effectiveness, match with expectations and engagement result higher at the post-intervention assessment after receiving the SH+ than after receiving the alternative intervention."
Sentence unclear to me. A comma after "with expectations" could make it clearer but I still think that "result higher" is then not correct.

14) p. 26 "These factors could be represented by the provision of structured and interactive activities, contents that stimulate reflection and expression of emotions, guided exercises and the stimulation of multiple senses through visual and auditory channels"
If possible a references/ references would be good as these are all empirical claims for causal connections.

15) p. 28: "Further, despite the fact that the drop-out rates were very similar between groups, proving intervention feasibility, we were not able to identify and analyse reasons related to the dropouts."
Similar question as above: no results are reported to support this statement and I question therefore whether it is admissible.

16) Typo in reference [22] "psychological"

17) Reference 58 ([58] Baum, C., Schaffer, M. (2009) Implementing econometrics estimators with Mata. Stata Users Group, United Kingdom Stata Users Group Meetings 2009) is a reference for the particular routine used in the analysis; for the introduction of SUR models probably a different resource should be provided.

Personal observation

I was surprised that the team does neither comment on the large (albeit anticipated) drop-out rate as well as the secular trend. I mention them both, as they may be interrelated issues.

Taking the observed data on their own, both groups are 'getting better' over time. One of the reasons for the lack of an effect could be that the assumption of a stable comparator group was violated as both groups changed substantially – and the added effect of the intervention is not strong enough.

But relatedly, this trend offers the potential for systematic drop-out, i.e. participants already more severely affected could be dropping out (in both groups admittedly at a similar rate), therefore shifting the group distributions (making the control group a less stable comparator and overestimating change in the intervention group).

As the paper presents baseline controlled analyses as a sensitivity and as long as one neither assumes differential drop out nor differential intervention effects, the impact on the cross-sectional pre-planned between-group comparisons should be minimal (under these assumptions we would still expect the same effect size). I nevertheless thought I just share this observation as it is a fairly typical problem in trials and no mention of this was found (see also (15) above).

Decision letter (RSOS-210219.R2)

Dear Professor Rusconi:

On behalf of the Editor, I am pleased to inform you that your Stage 2 Registered Report RSOS-210219.R2 entitled "EFFECTIVENESS OF SELF-HELP PLUS (SH+) IN REDUCING ANXIETY AND POST-TRAUMATIC SYMPTOMATOLOGY AMONG CARE HOME WORKERS IN THE COVID-19 PANDEMIC: AN RCT" has been deemed suitable for publication in Royal Society Open Science subject to minor revision in accordance with the referee suggestions. Please find the referees' comments at the end of this email.

The reviewers and Subject Editor have recommended publication, but also suggest some minor revisions to your manuscript. We invite you to respond to the comments and revise your manuscript. Below the referees' and Editors' comments (where applicable) we provide additional requirements. Final acceptance of your manuscript is dependent on these requirements being met. We provide guidance below to help you prepare your revision.

Please submit your revised manuscript and required files (see below) no later than 7 days from today's (ie 29-Oct-2021) date. Note: the ScholarOne system will 'lock' if submission of the revision is attempted 7 or more days after the deadline. If you do not think you will be able to meet this deadline please contact the editorial office immediately.

Please also be aware that we require all authors to have an active email address at the time of submission and acceptance, but note that chiara.bove@studenti.unitn.it is currently not receiving emails from the journal. Please can you either check that this email address is correct, ensure emails from the journal are 'white-listed' by the relevant IT providers, or supply an alternative email address at revision?

Please note article processing charges apply to papers accepted for publication in Royal Society Open Science (<https://royalsocietypublishing.org/rsos/charges>). Charges will also apply to papers transferred to the journal from other Royal Society Publishing journals, as well as papers submitted as part of our collaboration with the Royal Society of Chemistry

(<https://royalsocietypublishing.org/rsos/chemistry>). Fee waivers are available but must be requested when you submit your revision (<https://royalsocietypublishing.org/rsos/waivers>).

on behalf of Professor Chris Chambers
(Registered Reports Editor, Royal Society Open Science)
openscience@royalsociety.org

Associate Editor Comments to Author (Professor Chris Chambers):

Associate Editor: 1

Comments to the Author:

Two of the three reviewers from Stage 1 were available to evaluate the completed Stage 2 manuscript. As you will see, the comments are broadly very positive while noting several areas that would benefit from clarification, principally in the reporting and interpretation of the results (e.g. justification of certain exploratory analyses; ensuring that confirmatory and exploratory outcomes are clearly distinguished; ensuring that the interpretation of exploratory analyses is appropriately evidence-bound; consideration of limitations in the Discussion). All issues appear to be readily addressable through a round of careful revision. Provided the authors are able to respond comprehensively to all concerns, a final Stage 2 decision should be forthcoming without requiring further in-depth review.

Comments to Author:

Reviewer: 1

Comments to the Author(s)

This paper presents the second stage of a registered report on the effectiveness of a self-help intervention (SH +) versus an active control in reducing anxiety and PTSD symptoms in care home workers during the COVID-19 pandemic. From the pre-registered primary analysis, the authors did not find evidence that the SH + intervention was more effective than the active control. This study makes an important contribution to the literature in testing the effectiveness of the SH + intervention in a randomised design with an active comparison group. Based on their findings the authors speculate that previously observed positive effects of the SH + in studies without control groups may be due to non-specific factors.

The authors have made minor changes to the introduction section. For the most part these changes were warranted in updating the text to reflect the current situation. The stated hypotheses are the same as the approved stage 1 submission, and the data are able to test the proposed hypotheses. The authors adhered to the registered experimental procedures except for a 2-week delay to the planned timepoint for the time 2 follow-up. The authors have been transparent about this change and note that it does not influence their primary pre-registered outcome.

I found the results section to be overly dense which made understanding the researcher's findings difficult. I think it would be beneficial to include some of the results in supplementary

materials. For example, the authors could note that controlling for imbalance at baseline did not impact their findings in the main manuscript and include the details of these analyses in supplementary materials. It would also be informative to highlight which analyses were registered versus exploratory in the results section.

The tables and figures were somewhat difficult to read in their present format. It would be helpful to either separate figure 3 into two separate figures depending on the outcome (% of mild symptoms and difference in percentage) or just focus on one outcome rather than have two y axes. Adding an additional row or column to highlight the timepoints in table 4 (as the authors have already done in table 5) would be helpful. Using more descriptive labels for the timepoints in the tables and figures would also help clarify the author's findings (e.g., baseline, post-intervention and 14-week follow-up rather than T0, T1, T2).

I have not previously come across the Cochran's omnibus Q test conducted by the authors and am therefore not able to comment on whether this unregistered exploratory analysis was statistically justified or methodologically sound. However, at present I feel that this test is not informative in the way results are presented. I found it confusing that the authors report significant findings for this analysis except for between T1 and T2 in the alternative group, but do not provide any further comment on this. The descriptive statistics seem to indicate that the whereas the SH+ group showed a decline from T1 to T2 in mild symptoms in at least 1 scale (74.12% to 64.71%; Table 4) the alternative group showed relatively stable levels (76.47% to 74.67%) which would seem to indicate some benefit of the SH+ in the long-term. It may be that I am misinterpreting these findings, but I believe further commentary would be helpful in helping readers to understand the practical significance of this analysis if it warrants inclusion as an exploratory test. Aside from this I feel that the author's conclusions are justified given the data.

Reviewer: 3

Comments to the Author(s)

I thank the editorial team for inviting me to review the Stage 2 version of this paper and I congratulate the team to finish their study so closely within the confines of their registration.

I can confirm that the Introduction, rationale and stated hypotheses are the same as the approved Stage 1 submission. The report adheres to the registered experimental procedures and minimal deviations from the original approach are transparently described and in my view largely (see comments below) within the affordances of such a practice-based research project.

The presented data are able to test the authors' proposed hypotheses and the exploratory analyses are clearly separated in the report. The authors' conclusions are largely justified given the data and reported results (comments on smaller aspects below).

I provide my comments below separated into Major and Minor points as well as a personal observation relating to the results – the latter clearly separated as it may invite what feels like undue speculation on the results and is not a critique based on comparisons with the IPA manuscript.

__MAJOR__

1) Page 19, " Finally, only one (the first to enrol) of any two or more responding workers from NCH who shared their accommodation with one another were entered into the randomization." This is a difficult point that would require at least further discussion as a limitation. This detail of the procedure was not registered – in the Stage 1 manuscript it was only specified that: p. 7: "Those meeting the inclusion criteria outlined below, completing an initial survey (demographic information and baseline assessment) and passing the validity check at the

baseline assessment (see section 2.4) will then be randomly assigned to one of the two activity groups. Randomization will be stratified by centre and the RCT will be conducted in accordance with the Consolidated Standards of Reporting Trials statement."

And it was later specified, p. 8, that:

"To avoid contamination, only one person per household will be randomised."

As conducted and reported, in study planning this should have been an inclusion criterion i.e. stating that the first to enrol from shared accommodation will be randomised.

As conducted now, this is effectively a randomisation breach as the protocol/Stage 1 paper defined everyone as randomizable.

As far as I understand, the likely effect for this trial will be minimal as the authors report on page 19 "This led to exclusion of a further two (N = 2) respondents from randomization (see Figure 2)." Nevertheless, this should at least be noted.

2) p. 27: The discussion of secondary results starting with "In exploratory analyses,..." seems to go a bit far when compared to the registration as well as to the results that are reported in the manuscript.

2a) "Since the SH+ has proved to be predominant for these characteristics, even in a conservative test like ours (albeit in secondary analyses), we can hypothesise that its use might be effective for skilled healthcare staff working in long-term healthcare facilities beyond emergency situations."

2a-i) It is unclear to me what "predominant" means in this case and how this relates to the strong interpretation offered.

2a-ii) It is unclear to me which test is described as 'conservative' here. From a statistical perspective any analysis this can refer to would seem to be pretty liberal as it was not pre-registered.

2b) "Any further speculation on these findings would be unwarranted,..."

It is unclear to me why this small paragraph was included as it contains such 'further speculation'. Especially, the link to stepped care models seems to be a long shot as the present study was not conducted in such a context.

3) p. 28: " Moreover, we provided a large sample of workers in NCH with a practical strategy to independently manage distress, which was perceived - according to secondary analyses - as more effective and engaging than an alternative but similarly demanding activity. If this was confirmed, the strategy could be adapted for many other groups, which is particularly important given the potential long-term consequences of the COVID-19 pandemic."

This interpretation seems to assign undue relevance to the results of analyses that were not pre-planned as the primary outcome. This should either be dropped from the manuscript or the wording revised to reflect the exploratory nature of the results.

4) p. 28 " Exposure to chronic stressors and multiple traumatic experiences may generate long-term consequences even after the acute phase of the pandemic, especially for workers in NCH." I am not sure why this sentence is placed in the "strengths" section.

If kept, there should be a reference provided - the presented empirical statement does not follow from this study.

5) Table 5 and its description are difficult to match to the description of the methods applied in 2.7.3.2. If I interpreted correctly that these are the results connected to that part of the methods section, the linkage in terminology could be increased (also: the SUR results which are indicated as the first methodological step seem not to be reported).

__MINOR__

6) What is the RSOS convention for decimal places as a number of different formats are used.

7) p. 22, "The Cochran's test turned out to be statistically significant..."
Suggestion: "was statistically significant"

8) p. 24: "Joint analyses on the GAD-7 and IES-R scores controlling for baseline values did not show any statistical significance (p-value 0.544 at T1 and 0.254 at T2) in the SUR equation model."
Partly a question to the editor: Is it admissible to report results without reporting full results (coefficients, SEs, and in case of SUR especially the residual correlations)?

9) p. 28: "...the blindness of participants and of the majority of the researchers involved..."
I suggest to reformulate, as neither participants and researchers were physically blind (I presume...)

10) Table 2 is not introduced/linked in the text.

11) Table 4 contains drop-out ratios, which seem not to be commented on/explained in the text accompanying the table.

12) Page 18, "Number of log-ins was dic[h]othomized (no log-ins vs. at least one log-in),..."
Typo as indicated and it is not clear why the distribution of the raw data is not presented as well.

13) p. 26 "In exploratory analyses, perceived intervention effectiveness, match with expectations and engagement result higher at the post-intervention assessment after receiving the SH+ than after receiving the alternative intervention."
Sentence unclear to me. A comma after "with expectations" could make it clearer but I still think that "result higher" is then not correct.

14) p. 26 "These factors could be represented by the provision of structured and interactive activities, contents that stimulate reflection and expression of emotions, guided exercises and the stimulation of multiple senses through visual and auditory channels"
If possible a references/ references would be good as these are all empirical claims for causal connections.

15) p. 28: "Further, despite the fact that the drop-out rates were very similar between groups, proving intervention feasibility, we were not able to identify and analyse reasons related to the dropouts."
Similar question as above: no results are reported to support this statement and I question therefore whether it is admissible.

16) Typo in reference [22] "psychological"

17) Reference 58 ([58] Baum, C., Schaffer, M. (2009) Implementing econometrics estimators with Mata. Stata Users Group, United Kingdom Stata Users Group Meetings 2009) is a reference for the particular routine used in the analysis; for the introduction of SUR models probably a different resource should be provided.

__Personal observation__

I was surprised that the team does neither comment on the large (albeit anticipated) drop-out rate as well as the secular trend. I mention them both, as they may be interrelated issues. Taking the observed data on their own, both groups are 'getting better' over time. One of the reasons for the lack of an effect could be that the assumption of a stable comparator group was

violated as both groups changed substantially – and the added effect of the intervention is not strong enough.

But relatedly, this trend offers the potential for systematic drop-out, i.e. participants already more severely affected could be dropping out (in both groups admittedly at a similar rate), therefore shifting the group distributions (making the control group a less stable comparator and overestimating change in the intervention group).

As the paper presents baseline controlled analyses as a sensitivity and as long as one neither assumes differential drop out nor differential intervention effects, the impact on the cross-sectional pre-planned between-group comparisons should be minimal (under these assumptions we would still expect the same effect size). I nevertheless thought I just share this observation as it is a fairly typical problem in trials and no mention of this was found (see also (15) above).

===PREPARING YOUR MANUSCRIPT===

one version should clearly identify all the changes that have been made (for instance, in coloured highlight, in bold text, or tracked changes);

===PREPARING YOUR REVISION IN SCHOLARONE===

-- If you are requesting an article processing charge waiver, you must select the relevant waiver option (if requesting a discretionary waiver, the form should have been uploaded, see 'File upload' above).

-- If you have uploaded any electronic supplementary (ESM) files, please ensure you follow the guidance at <https://royalsociety.org/journals/authors/author-guidelines/#supplementary-material> to include a suitable title and informative caption. An example of appropriate titling and captioning may be found at https://figshare.com/articles/Table_S2_from_Is_there_a_trade-off_between_peak_performance_and_performance_breadth_across_temperatures_for_aerobic_scope_in_teleost_fishes_/3843624.

Author's Response to Decision Letter for (RSOS-210219.R2)

See Appendix E.

Decision letter (RSOS-210219.R3)

Dear Professor Rusconi:

It is a pleasure to accept your manuscript entitled "EFFECTIVENESS OF SELF-HELP PLUS (SH+) IN REDUCING ANXIETY AND POST-TRAUMATIC SYMPTOMATOLOGY AMONG CARE HOME WORKERS IN THE COVID-19 PANDEMIC: AN RCT" in its current form for publication in Royal Society Open Science.

COVID-19 rapid publication process:

We are taking steps to expedite the publication of research relevant to the pandemic. If you wish, you can opt to have your paper published as soon as it is ready, rather than waiting for it to be published the scheduled Wednesday.

This means your paper will not be included in the weekly media round-up which the Society sends to journalists ahead of publication. However, it will still appear in the COVID-19 Publishing Collection which journalists will be directed to each week (<https://royalsocietypublishing.org/topic/special-collections/novel-coronavirus-outbreak>).

If you wish to have your paper considered for immediate publication, or to discuss further, please notify openscience_proofs@royalsociety.org and press@royalsociety.org when you respond to this email.

Please see the Royal Society Publishing guidance on how you may share your accepted author manuscript at <https://royalsociety.org/journals/ethics-policies/media-embargo/>. After publication, some additional ways to effectively promote your article can also be found here

<https://royalsociety.org/blog/2020/07/promoting-your-latest-paper-and-tracking-your-results/>.

Thank you for your fine contribution. On behalf of the Editors of Royal Society Open Science, we look forward to your continued contributions to the journal.

on behalf of Professor Chris Chambers (Subject Editor)
openscience@royalsociety.org

Appendix A

Peer review of RSOS RCT for SH+

Summary of the study

The proposed study examines the use of self-help plus (SH+) to reduce anxiety in workers in nursing and care homes during COVID-19 via a randomised controlled trial, which will be undertaken in Northern Italy. The comparator is 'active' in the sense that it is an intervention that may also be expected to benefit participants, which includes reading and guided reflection. The data analyst will be masked to the allocation and participants will be masked to the study aims and number of groups. The primary outcome is composite, comprising the proportion of individuals reporting anxiety and/or post-traumatic symptomology at 1-week post intervention.

The authors are attempting to study an important question, and have shared materials along with their submission, including code for sample size calculations, which should be commended. They have specified the statistical test that will be used for the primary outcome.

It is worth noting that recruitment to the study has already begun. It is not clear if this means that changes to the trial plan following this review would not be possible (e.g. if the participants have already consented to the study and changing it would mean that recruitment has to restart). This should be clarified.

Note that page numbers in my review refer to the numbers page numbers used in the upper right corner of the protocol, not the pdf page number (so abstract is page 2).

Overall I think the study will be valuable, though I have a number of suggestions for improvements.

Specific questions posed by the journal

1. The scientific validity of the research question(s)

The research question appears to be valid.

2. The logic, rationale, and plausibility of the proposed hypotheses

Below I ask for justification of the expected effect size. I also outline some concerns about the control arm. If addressed, I think the study will meet these criteria.

3. The soundness and feasibility of the methodology and analysis pipeline (including statistical power analysis where applicable)

I have made suggestions for improvements. The power analysis appears to be acceptable (though I do have comments on how it might be changed) and I have checked the calculation.

4. Whether the clarity and degree of methodological detail would be sufficient to replicate exactly the proposed experimental procedures and analysis pipeline

More details are needed on the intervention and facilitator role, in particular. I also provide other suggestions for improvements in clarity and detail, e.g. around recruitment and randomisation. Improving clarity and detail I think is a major area for improvement.

5. Whether the authors provide a sufficiently clear and detailed description of the methods to prevent undisclosed flexibility in the experimental procedures or analysis pipeline

I do not think that flexibility in the experimental procedure or analysis will be problematic provided extra details are provided as requested throughout.

6. Whether the authors have considered sufficient outcome-neutral conditions (e.g. positive controls) for ensuring that the results obtained are able to test the stated hypotheses

They have defined some validity checks which seem reasonable to me. I have made some non-essential suggestions for further checks that might be useful.

General comments

The purpose of the study and its implications could be better described. For example, if the intervention is shown to be effective, will it be deployed for use? Or is this study intended to show that in principle the intervention can provide a benefit? This is important because, if the former, the intervention needs to represent the intervention that would be used in the 'real world'.

In terms of the primary outcome, have the authors considered an analysis as a change from baseline rather than simply overall proportion above a threshold in each group? I am not an expert in this and leave judgement to them, but such an approach may be better if there are imbalances in baseline assessment, and would give a more interpretable measure of the effect (e.g. an average change of 4 points on the GAD-7). The composite endpoint chosen will be difficult to interpret and will not give a good indication of the magnitude of the effect for each individual (for example, there may be a reduction from 0.8 to 0.6 overall, and that might mean everyone moved 2 points on a scale, or they could have moved 10 points...). If they decide to use this intervention in practice following a successful trial, it would be very hard to explain to workers what the effects are likely to be for them, and this limits the utility of the study in my opinion.

The authors do not include details of the secondary analysis plans and they justify this decision: "As the secondary outcomes are included for exploratory purposes, only the hypothesis and analysis plan for the main outcome will be preregistered with the protocol, in compliance with the Registered Report format requirements." However, it is conventional for these to be prespecified and certainly to appear in the methods section of a paper. Personally I think it would be better to include details on how these will be analysed at this stage but this can be decided by the study team/editor.

I struggled to follow the recruitment process and how the required sample size will be achieved. Will all workers in participating NCH be contacted at once, and all of those that wish to be will be assessed for eligibility? If many more than 164 are eligible, will all still be included in the trial? Since the primary analysis is (effectively) intention to treat but will exclude some participants after randomisation, I think the target sample size needs to be adjusted up to account for potential drop out due to participants never accessing the intervention.

There is also the potential for loss to follow-up after randomisation and after accessing the intervention. How will those participants be treated in analysis? Some loss to follow-up should be assumed and adjusted for in the sample size target if appropriate. Treatment of missing outcome data in general needs to be described.

The facilitator represents a potential source of bias in the study, in that they are aware of the intervention assignment and interact with the participants: as such, they could treat participants in different groups differently (and could plausibly favour the intervention). This should be further discussed and mitigated if possible. Would it not be possible to simply send all participants a message/email reminder once per week, rather than interacting with them more substantially? The fact the usefulness of the facilitator will be rated may provide a check on whether or not the facilitator treated the groups equally (this could be added as an additional validity check).

Regardless, the role of the facilitator needs to be explained more fully. The text states “The alternative activity is designed to 1) partial out improvements that could be elicited by facilitator attention, individual engagement and expectations rather than by specific content”. Is the facilitator not part of the SH+ intervention? If they are, then it’s not clear why the effect of the facilitator needs to be differentiated from the audio and reading material. If the facilitator is not part of the intervention, then why is a facilitator included at all (as when the intervention is used in a real world setting there would not be one)?

Details of the facilitator’s participation and training should also be clarified. What will the facilitator message the participants? What will the facilitator say on the call in the case of no response? What training will the facilitator be given, and can this training material be shared so that the experiment can be repeated and so that the intervention can be used in practice in the same way it was tested if it shown to work?

I would like to know more about the risk of contamination of the intervention amongst participants within study centres. Is it possible that participants will discuss the intervention amongst themselves and become unblinded to the study arms? Perhaps estimating the proportion of staff in each centre that you are expecting to recruit to the study would be useful in assessing this.

Before commencing the trial, the authors should register their study on a clinical trial registry. That they will do this and where and when should be stated in the protocol.

I have included a CONSORT checklist with the methods items completed according to my assessment of the article and have noted items where I’ve requested more details (because the authors stated they would do the study according to CONSORT). It would be helpful if the authors could submit an updated checklist with their next submission and denote the page numbers where each item is reported (or an explanation of why it is not applicable). Obviously this is at the discretion of the editor and team but would certainly be useful. They may also wish to consult the SPIRIT statement for RCT protocols.

Specific comments

Title: the title should state that it is an RCT as per CONSORT.

Introduction

Pg3 line 36: the acronym ACT should be defined.

Pg3 lines 44-52 discuss another trial of SH+ in a different setting and population. It would be useful to summarise the results of the trial in more detail: for example the magnitude of the effects found and the specific endpoints used. The primary endpoint used in that study (K6 score) differs from that used in this study – the reasoning for using a different endpoint could be described. That trial used considerably more participants than the present study and it may be worth commenting on why in the current trial the sample size is much smaller.

It would be useful to note whether the RCT mentioned on line 44 is the only RCT of SH+ currently available and, if there are others, to describe them. Clinical trial registries (e.g. clinicaltrials.gov or ICTRP) could be briefly searched to determine whether there are any other ongoing studies of the same intervention.

Pg3 line 53 to next page describe the intervention and control arm. The rationale for using an active comparator is given, but the specific control chosen is not well justified: the authors are creating an intervention, but is there no established method with demonstrated or possible effect that SH+ could be compared against? If there are no established methods, then a preferred control arm might more closely approximate usual care (e.g. in Tol et al 2020, ref 16 they use ‘enhanced usual care’) and in registered studies found on clinicaltrials.gov (NCT03571347 and NCT03587896) use enhanced usual care. Since nothing is known about the efficacy of the proposed control arm, a finding of no difference between the groups would be difficult to interpret – both arms could be effective, or both ineffective, and as such no useful finding generated. I do appreciate that changing this may not be possible due to the requests from the NCH etc.

Pg 4 lines 52-54: “as a psychological intervention to increase the ability of nursing and care home workers to cope with stressful situations during the second wave of COVID-19 contagions”. It may be that we are no longer in the second wave of the pandemic by the time the trial is completed and the intervention can be used, so using a less restrictive term could be better (e.g. “during the pandemic”).

Materials and methods

Pg5 lines 38 – 44: this information is background info and is not relevant to the sample size calculation (the subheading it is under).

Pg5 lines 57-60: “Also, whatever the current prevalence, the SH+ intervention will provide participants with very simple but effective tools to cope with an acute or chronic stressful situation at work.” We do not yet know if the SH+ intervention will be effective. Also, prevalence in each group influences the required sample size. This statement should be removed in my opinion.

Pg6 lines 8 – 12: Why is a relative risk reduction of 25% expected? How does this compare to effect sizes of other studies of psychological distress in general, and specifically to the other SH+ trial mentioned in the introduction.

Pg6 lines 13 – 14: power of 80% is conventional and acceptable (so does not have to be revised). However, since the intervention is probably quite cheap to deliver, I did wonder if

the authors had considered using a higher power (e.g. 0.9 which would require ~106 participants per groups) as 20% is quite a high false negative rate. Below I note that a one-sided test may be appropriate which would increase power for this sample size if you choose to do it.

Pg6 lines 16 – 19: “Since we cannot provide an accurate prediction of the attrition/non-compliance rate with this population (in our previous epidemiological study, the rate was 5% but the current study includes two crucial data locks for the preregistered analysis)”: which study is “our previous epidemiological study” referring to? It should be cited here. If it is Riello et al 2020, I do not see how any information in that study is relevant to non-compliance (since there was no intervention) so the sentence should be reworded.

Pg 6: “Randomization will be stratified by centre and the RCT will be conducted in accordance with the Consolidated Standards of Reporting Trials statement” – CONSORT is a reporting guideline, not a guideline for conduct of RCTs. I would change the terminology to be “reported according to CONSORT” (and then ensure that it is reported according to consort). This statement is not relevant to recruitment and inclusion criteria (the section it’s in) so may be better placed at e.g. the beginning of the methods.

Pg6 lines 30-35: in recruitment and RCT inclusion criteria, the centres where recruitment will and has taken place should be listed and the prevalence study should be referenced when mentioned.

Pg6 lines 46-55: are there any exclusion criteria? In the other SH+ trial (Tol et al 2020) they state that “Self-Help Plus is not intended for people with complex mental health problems (such as psychosis) or those at imminent risk of suicide.”. Should this be addressed in the present study?

Pg7 line 7: “To avoid contamination, only one person per household will be randomised.” How will the study staff know whether someone from the same household has already been randomised? I could not see a question in the OSF material about this.

Pg7 line 13: “the randomization list will be accessible only to the data manager and to the website manager, for the assignment of activities to individual participants”. Which of the website manager and data manager will assign participants to interventions? Both? Why?

Pg7 lines 5-23: Will the individuals making assigning participants have any access to information about baseline assessment? If not, could you please describe how they will be unaware of this information (e.g. they will simply receive the email addresses of eligible participants, randomly ordered).

Pg7, lines 51:56 and Figure: this information is not specific to the characteristics of the intervention (the subheading). I would suggest it is moved to the start of the methods where it would provide a nice overview of the planned approach.

Pg8 line 11 states that the activities will be disclosed to participants on week 3 of the RCT. From the figure I would have thought they'd be disclosed on week 2, when the intervention begins. Please clarify.

Pg8 SH+ description. For clarity, I would suggest that this section be dedicated to describing the actual intervention that will be used in the study, rather than background information on it which could be included in the introduction (i.e. lines 29-49 could be elsewhere as they don't describe the intervention given to the participants).

A sample of materials is provided, including very helpfully in English. However, the exact method of delivery and full details of the intervention and control are not provided. A full description and preferably materials is essential to allow others to reproduce the study or to use the intervention in the event that SH+ is found to be efficacious. As I understand it, participants will access the information through a website where they need to log in. Can this website and the intervention in its usable form be shared? If not, it should all be described in more detail. Other details like how long the intervention takes and the frequency with which the participants will be expected to engage with it should be given. This comment applies both to the intervention and control arm.

Page 11- 13: there are a lot of different scales that participants are expected to complete. How much time are these likely to add to assessments for participants? Have the authors considered that this might put off some participants and reduce completion rates?

Notwithstanding the point about overburdening participants, could it be worth collecting information on reasons for dropout/lack of engagement with the intervention? This seems potentially more valuable to me than the different measures of well-being and anxiety.

Page 12 table 1: the timing of the post-intervention assessment (column 3) should be clarified. Based on Figure 1 I think it means 1 week.

Page 13 validity checks: it might be useful to list the validity checks and the specific criteria that will be considered as 'valid' vs. not so that it will be clear in the results whether the checks were passed. You could consider adding something about level of engagement in the materials – for example, if all the participants in the SH+ arm log in once to the materials but do not use them after that, this might indicate there is some issue with the delivery of that specific intervention via its website. You could also ask someone to test the materials (i.e. access and quickly go through them) in the format they will be used by participants to check for obvious software issues. You could also collect data on what participants liked and didn't like about the intervention to inform future interventions and give an opportunity to highlight major issues with its delivery.

Page13 line 32-35: if more than one response is sent from one device, what will be done?

Page 14, table 2: why is the analysis two-tailed? The hypothesis is framed as a reduction in proportion in the intervention group (rather than a difference in either direction), which seems sensible to me as it seems unlikely that an increase in proportion in the intervention group vs control would be observed. Using a one tailed test would give more power (for the

same sample size) and would I think test the stated hypothesis. Note that if this is changed the power calculation needs to be updated.

Page 15, lines 41 - 44: generally, it is not a good idea to rely solely on the p-value for the primary outcome. Can the authors include a measure of precision of the estimated effect size (e.g. 95% CI) or explain why this is not appropriate? This will be needed as part of the CONSORT statement when reporting results and it is best to prespecify it.

Appendix B

CONSORT 2010 checklist of information to include when reporting a randomised trial*

Section/Topic	Item No	Checklist item	Reported on page No
Title and abstract			
	1a	Identification as a randomised trial in the title	Requested
	1b	Structured summary of trial design, methods, results, and conclusions (for specific guidance see CONSORT for abstracts)	NA at this stage
Introduction			
Background and objectives	2a	Scientific background and explanation of rationale	2-3
	2b	Specific objectives or hypotheses	4-5
Methods			
Trial design	3a	Description of trial design (such as parallel, factorial) including allocation ratio	7
	3b	Important changes to methods after trial commencement (such as eligibility criteria), with reasons	NA
Participants	4a	Eligibility criteria for participants	6
	4b	Settings and locations where the data were collected	More detail requested
Interventions	5	The interventions for each group with sufficient details to allow replication, including how and when they were actually administered	More detail requested
Outcomes	6a	Completely defined pre-specified primary and secondary outcome measures, including how and when they were assessed	Secondary outcomes not included currently (but justified)
	6b	Any changes to trial outcomes after the trial commenced, with reasons	NA
Sample size	7a	How sample size was determined	5
	7b	When applicable, explanation of any interim analyses and stopping guidelines	NA
Randomisation:			
Sequence generation	8a	Method used to generate the random allocation sequence	7
	8b	Type of randomisation; details of any restriction (such as blocking and block size)	7
Allocation	9	Mechanism used to implement the random allocation sequence (such as sequentially numbered containers),	More detail

concealment mechanism		describing any steps taken to conceal the sequence until interventions were assigned	requested.
Implementation	10	Who generated the random allocation sequence, who enrolled participants, and who assigned participants to interventions	More detail requested
Blinding	11a	If done, who was blinded after assignment to interventions (for example, participants, care providers, those assessing outcomes) and how	14
	11b	If relevant, description of the similarity of interventions	9
Statistical methods	12a	Statistical methods used to compare groups for primary and secondary outcomes	Primary provided but no method for precision, secondary not provided
	12b	Methods for additional analyses, such as subgroup analyses and adjusted analyses	Not provided
Results			
Participant flow (a diagram is strongly recommended)	13a	For each group, the numbers of participants who were randomly assigned, received intended treatment, and were analysed for the primary outcome	
	13b	For each group, losses and exclusions after randomisation, together with reasons	
Recruitment	14a	Dates defining the periods of recruitment and follow-up	
	14b	Why the trial ended or was stopped	
Baseline data	15	A table showing baseline demographic and clinical characteristics for each group	
Numbers analysed	16	For each group, number of participants (denominator) included in each analysis and whether the analysis was by original assigned groups	
Outcomes and estimation	17a	For each primary and secondary outcome, results for each group, and the estimated effect size and its precision (such as 95% confidence interval)	
	17b	For binary outcomes, presentation of both absolute and relative effect sizes is recommended	
Ancillary analyses	18	Results of any other analyses performed, including subgroup analyses and adjusted analyses, distinguishing pre-specified from exploratory	
Harms	19	All important harms or unintended effects in each group (for specific guidance see CONSORT for harms)	
Discussion			
Limitations	20	Trial limitations, addressing sources of potential bias, imprecision, and, if relevant, multiplicity of analyses	
Generalisability	21	Generalisability (external validity, applicability) of the trial findings	
Interpretation	22	Interpretation consistent with results, balancing benefits and harms, and considering other relevant evidence	

Other information

Registration	23	Registration number and name of trial registry	_____
Protocol	24	Where the full trial protocol can be accessed, if available	_____
Funding	25	Sources of funding and other support (such as supply of drugs), role of funders	_____

*We strongly recommend reading this statement in conjunction with the CONSORT 2010 Explanation and Elaboration for important clarifications on all the items. If relevant, we also recommend reading CONSORT extensions for cluster randomised trials, non-inferiority and equivalence trials, non-pharmacological treatments, herbal interventions, and pragmatic trials. Additional extensions are forthcoming: for those and for up to date references relevant to this checklist, see www.consort-statement.org.

ASSOCIATE EDITOR (AE)

AE#1: Four reviewers have now assessed the manuscript, including two field experts (Revs 1 and 3), one expert in trial methodology (Rev 2) and a specialist statistical editor. The reviews overall are cautiously positive. Key issues to address in revision include clarification and justification of key elements of the methodology and rationale, addressing contamination effects, concerns regarding variable dichotomization (raised by multiple reviewers), interpretation of non-significant results (and potential alternative analyses to consider), sufficiency of positive controls, and adherence to trial reporting standards.

Reply to AE#1: *We are very grateful to the Editors and the Reviewers for their positive assessment. We have now addressed all of the points raised by either justifying our choices or making changes to our protocol, where possible, as detailed in our replies below.*

AE#2: Concerning the point raised by reviewer 2: "Before commencing the trial, the authors should register their study on a clinical trial registry. That they will do this and where and when should be stated in the protocol." Note that as part of this RSOS special initiative, the journal preregisters the Stage 1 protocol at the point of in-principle acceptance, and we will follow up with you in due course about that.

Reply to AE#2: *Thank you for confirming this, we have now received the checklist for preregistration and returned it completed, in parallel with this resubmission.*

AE#3: Concerning the prespecification of secondary outcomes and analyses, I am open to these being included in the Stage 1 manuscript (as suggested by several reviewers — and there are good arguments for doing so in the context of a clinical trial); alternatively, provided these analyses are going to be strictly exploratory in nature (and not dominate the conclusions) then the authors are welcome to reserve mention of these analyses until Stage 2. This approach differs somewhat from expectations in conventional trial reporting.

Reply to AE#3: *Thank you for clarifying this point. We have now followed the Reviewers' suggestion and prespecified the secondary analyses we intend to perform on primary and secondary outcome measures in our Stage 1 manuscript.*

SPECIALIST STATISTICAL EDITOR (SSE)

SSE#1: The authors use what is meant to be an active control, to control for in their words, expectation, facilitator attention, and participant engagement. Outcome neutral tests are needed for each of these claims. For example, for expectation, they could ask participants to give their pretest scores and predict their posttest scores on the outcome DVs, after reading the materials of the intervention, but before the intervention. They could instead run a separate expectation elicitation experiment, where subjects read the materials of each intervention and give the ratings as above - the difference being one can collect expectations before and separate to the main experiment.

Reply to SSE#1: *In our protocol we will be able to ensure that assistant (former facilitator) attention is objectively the same, by keeping constant our assistant's interactions between the groups (e.g. same type and number of reminders sent per participant in each group), and to measure perception of assistant attention, by asking each participant to rate how helpful the assistant had been to them. We will also be able to check participants' engagement objectively, by tracking their activity on the website (e.g. number of logins to the website), and subjectively, by asking them to rate both how engaging they found the intervention and whether they had been able to engage in the activities as much as they would have liked (indeed the intervention might have looked engaging but lack of time or unforeseen events might have prevented them from engaging as much as they would have liked to engage). These two questions have now been added to the protocol at the post-intervention assessments (please see highlighted questions in the Outcome measures folder on the OSF page). We will also induce similar expectations in the two groups at the informed consent stage, before participants can access the materials, but we will not be able to ask our participants to predict their post test scores on the outcome DVs after reading the materials, because reading the materials is part of the intervention itself. Due to ethical considerations it would also not be possible to run an ad hoc, adequately powered control experiment, with an independent sample from the same population of professionals during the emergency. Running it with university students would not be informative, as they are far from representative of our target population. The presence of an active control condition – as opposed to e.g. a “treatment as usual” condition – in our study enables the setting of similar expectations about the intervention between groups at the very outset. We will be able to check whether the information released to participants induced similar expectations in the two groups by obtaining a measure of expectations before participants can access the assigned materials, and we have now included a question to this effect at the end of the baseline assessment (please see highlighted questions in the Outcome measures folder on the OSF page). However, since this will be presented as an intervention study to improve psychological wellbeing (as this will exclude the potential stigma associated with participation in a trial on anxiety and PTSD symptoms), we will only check expectations concerning general psychological wellbeing and consider these as a proxy for expectations on anxiety and PTSD symptoms. The materials and activities for our active control are designed in such a way as to mimic the structure of the main intervention, and their contents have been chosen to look plausible for an intervention approach. At the end of the intervention, we will be able to check whether the proposed intervention matched our participants' initial expectations concerning their psychological wellbeing. Since this will be confounded with the perceived efficacy of the intervention, we have now added two questions at the post intervention assessments: one about the fulfilment of expectations, another about perceived efficacy of the intervention (please see highlighted questions in the Outcome measures folder on the OSF page).*

SSE#2: Equivalence of expectations could be tested by a Bayes factor where the obtained effect on the crucial outcome variables is used to inform what expectation might explain that outcome. The simplest hypothesis could use the obtained effect as the SD of a half-normal on the expected effects.

Reply to SSE#2: *Variables related to expectations will be compared across the two groups. In case of imbalance (as measured by a SMD above 0.1 in absolute value), such variables will be included in secondary analyses having the scale scores as outcome.*

SSE#3: Using a cut off on the scales to make a binary division can considerably reduce power. Why not use the continuous scales themselves? Maybe clinical interest is in the cut offs, but if there is no strong case for using cut offs in terms of clinical interpretation, from a statistical point of view using continuous variables is better.

Reply to SSE#3: *Although we agree that dichotomization involves loss of information which may lead to reduction in efficiency in the statistical analysis, we wish to maintain a cut-off approach as reasoning in terms of changes in the number of people who potentially need treatment is more relevant from a clinical, organizational, and public health perspective rather than reasoning in terms of changes to individual means; in this study, we are interested in whether SH+ reduces the frequency of people with psychological distress, thus we need to determine if the risk differs between intervention and control groups. In addition, reasoning in terms of changes to individual means often leads to high uncertainty in terms of clinical relevance of the observed mean differences (Moncrieff and Kirsch, 2005).*

SSE#4: For power an effect of interest is plucked from the air. Where does this come from? Why not another effect? And as this is a predicted effect not a smallest effect of interest, it should be used for Bayes factors not power. Or else the authors should justify a smallest effect of interest to calculate power.

Reply to SSE #4: *The Editor is correctly pointing out an important issue, that was thoroughly discussed within the research team. The choice of 20% as the smallest effect of interest is based on the following arguments:*

1. *We analyzed prevention trials on similar psychological interventions - either SH+ (Purgato et al., 2019) or other web-based guided self-help interventions (Buntrock et al., 2016) - and using a dichotomous measure as a primary outcome. In these trials, the authors hypothesized an absolute risk reduction of at least 10% for the incidence of mental disorders. This threshold was derived after consultation with clinical experts and stakeholders, and at the end of the trial authors identified an absolute difference close to 15%. Additionally, we considered that available trials were focused on prevention, with participants not severely distressed at the beginning of the study. In those contexts, anticipating a difference higher than 10-15% would have had little clinical sense. In our study, considering the results of the survey by Riello et al (2020), in which more than 40% of participants were moderately or severely distressed, a more substantial clinical change might be introduced by the SH+ intervention. Therefore, to increase the clinical relevance of this study, we decided to aim to detect a substantial clinical effect associated with SH+, which was operationalised as a 20% difference.*

2. *Given that we are exploring the efficacy of an intervention that has the potential for a large scaling-up, we needed to identify a level of difference between arms that represents a good compromise between a statistical reasoning (power calculation) and a clinically sound reasoning. For this reason, and in consideration of the public health impact of this research, it is critical to consider an effect which is strong also from a practical/implementation perspective.*

References

Buntrock C, Ebert DD, Lehr D, Smit F, Riper H, Berking M, Cuijpers P. 2016 Effect of a Web-Based Guided Self-help Intervention for Prevention of Major Depression in Adults With Subthreshold Depression: A Randomized Clinical Trial. *JAMA*. May 3; **315**(17):1854-63. doi: 10.1001/jama.2016.4326.

Moncrieff J, Kirsch I. 2005 Efficacy of antidepressants in adults. *BMJ*. **331**(7509):155-157. doi:10.1136/bmj.331.7509.155

Purgato M, Carswell K, Acarturk C, Au T, Akbai S, Anttila M, Baumgartner J, Bailey D, Biondi M, Bird M, Churchill R, Eskici S, Hansen LJ, Heron P, Ilkkursun Z, Kilian R, Koesters M, Lantta T, Nosè M, Ostuzzi G,

Papola D, Popa M, Sijbrandij M, Tarsitani L, Tedeschi F, Turrini G, Uygun E, Välimäki MA, Wancata J, White R, Zanini E, Cuijpers P, Barbui C, Van Ommeren M. 2019 Effectiveness and cost-effectiveness of Self-Help Plus (SH+) for preventing mental disorders in refugees and asylum seekers in Europe and Turkey: study protocols for two randomised controlled trials. *BMJ Open*. May 14; **9**(5):e030259. doi: 10.1136/bmjopen-2019-030259.

Riello M, Purgato M, Bove C, MacTaggart D, Rusconi E 2020 Prevalence of post-traumatic symptomatology and anxiety among residential nursing and care home workers following the first COVID-19 outbreak in Northern Italy. RSOS <https://doi.org/10.1098/rsos.200880>

REVIEWER 1 (R1)

This is an interesting piece of research making use of the COVID-19 pandemic to evaluate an important research question with clinical implications. The hypotheses were clear, and the methodology is plausible to allow the hypothesis to be answered. The quality checks were sufficient. The methodology of the study was detailed and would allow further replication. A few minor comments are detailed below:

R1#1: The introduction could be more concise, there was a lot of detail that was later repeated in the methods section.

Reply to R1#1: *We have now eliminated some repetitions between the Introduction and the Methods section (as also suggested by R#2), while still offering a succinct overview of our design approach in the Introduction.*

R1#2: When discussing the ACT components of the intervention in the introduction it reads as if the authors are planning to test the active components of the intervention or test it against another active intervention (e.g. “The potential beneficial effects of SH+ on psychological symptoms may be related to its main core components [...] Accordingly, ACT might lead to broader substantial changes than other forms of psychological interventions regarding psychological functioning”). It would be helpful to rephrase this if this is not the author’s intention.

Reply to R1#2: *We do expect some generic benefits from participation in the control intervention, therefore we have maintained the paragraph as is.*

R1#3: Further justification of the anticipated effect would be helpful. Has a reduction of 20% been based on previous literature?

Reply to R1#3: *Please see our reply to SSE#4.*

R1#4: The exact sample size the authors intend to recruit is currently unclear. The authors report a sample size calculation of 82 participants per group after attrition, but are unclear what the expected attrition rate is. It would be best to estimate an attrition rate based on similar trials and account for this in recruitment to ensure the study is not underpowered.

Reply to R1#4: *We did estimate an attrition/non-compliance rate based both on our knowledge of the population and on fairly similar literature, in order to identify a recruitment target but, since no trial has been previously conducted with this population, we thought it would be more cogent for Reviewers if we committed to reaching the minimum number of participants after attrition, rather than reporting what can only be an informed guess in our Stage 1 manuscript. However, we also concur with the Reviewer that reporting our rationale on the recruitment target that would enable reaching the desired power would improve transparency. Therefore, we have now reported an estimate for attrition rate and clarified our minimum target number for recruitment, which corresponds to 105 participants per group (please see section 2.1.2).*

R1#5: The authors may consider registering some of the additional outcomes currently listed as exploratory as secondary analyses to increase confidence in their findings.

Reply to R1#5: *We appreciate the Reviewer’s suggestion and we have now pre-specified our planned*

analyses on secondary outcomes (please see section 1.2. and 2.7.3).

REVIEWER 2 (R2)

Summary of the study

The proposed study examines the use of self-help plus (SH+) to reduce anxiety in workers in nursing and care homes during COVID-19 via a randomised controlled trial, which will be undertaken in Northern Italy. The comparator is 'active' in the sense that it is an intervention that may also be expected to benefit participants, which includes reading and guided reflection. The data analyst will be masked to the allocation and participants will be masked to the study aims and number of groups. The primary outcome is composite, comprising the proportion of individuals reporting anxiety and/or post-traumatic symptomology at 1-week post intervention.

The authors are attempting to study an important question, and have shared materials along with their submission, including code for sample size calculations, which should be commended. They have specified the statistical test that will be used for the primary outcome.

R2#1: It is worth noting that recruitment to the study has already begun. It is not clear if this means that changes to the trial plan following this review would not be possible (e.g. if the participants have already consented to the study and changing it would mean that recruitment has to restart). This should be clarified.

Reply to R2#1: *Recruitment has started at the level of NCH, whereas individual participants will be contacted and recruited only after receiving IPA (please see section 2.1.3). We have thus been able to tweak the protocol based on Reviewers' comments.*

Note that page numbers in my review refer to the numbers page numbers used in the upper right corner of the protocol, not the pdf page number (so abstract is page 2). Overall I think the study will be valuable, though I have a number of suggestions for improvements.

Specific questions posed by the journal

1. The scientific validity of the research question(s)

The research question appears to be valid.

2. The logic, rationale, and plausibility of the proposed hypotheses

Below I ask for justification of the expected effect size. I also outline some concerns about the control arm. If addressed, I think the study will meet these criteria.

3. The soundness and feasibility of the methodology and analysis pipeline (including statistical power analysis where applicable)

I have made suggestions for improvements. The power analysis appears to be acceptable (though I do have comments on how it might be changed) and I have checked the calculation.

4. Whether the clarity and degree of methodological detail would be sufficient to replicate exactly the proposed experimental procedures and analysis pipeline

More details are needed on the intervention and facilitator role, in particular. I also provide other suggestions for improvements in clarity and detail, e.g. around recruitment and randomisation. Improving clarity and detail I think is a major area for improvement.

5. Whether the authors provide a sufficiently clear and detailed description of the methods to prevent undisclosed flexibility in the experimental procedures or analysis pipeline

I do not think that flexibility in the experimental procedure or analysis will be problematic provided extra details are provided as requested throughout.

6. Whether the authors have considered sufficient outcome-neutral conditions (e.g. positive controls) for ensuring that the results obtained are able to test the stated hypotheses

They have defined some validity checks which seem reasonable to me. I have made some non-essential suggestions for further checks that might be useful.

General comments

R2#2: The purpose of the study and its implications could be better described. For example, if the intervention is shown to be effective, will it be deployed for use? Or is this study intended to show that in principle the intervention can provide a benefit? This is important because, if the former, the intervention needs to represent the intervention that would be used in the 'real world'.

Reply to R2#2: *We have now added a concise clarification on the purpose and possible implications of the study in the Introduction, without pre-empting our Discussion section. We have also clarified that this trial is already in the 'real world' (as per NCH request) and, if the SH+ turns out to be more effective than the control activity, it could be immediately deployed on a wider scale and with the same materials, support and infrastructure (please see section 1.1).*

R2#3: In terms of the primary outcome, have the authors considered an analysis as a change from baseline rather than simply overall proportion above a threshold in each group? I am not an expert in this and leave judgement to them, but such an approach may be better if there are imbalances in baseline assessment, and would give a more interpretable measure of the effect (e.g. an average change of 4 points on the GAD-7). The composite endpoint chosen will be difficult to interpret and will not give a good indication of the magnitude of the effect for each individual (for example, there may be a reduction from 0.8 to 0.6 overall, and that might mean everyone moved 2 points on a scale, or they could have moved 10 points...). If they decide to use this intervention in practice following a successful trial, it would be very hard to explain to workers what the effects are likely to be for them, and this limits the utility of the study in my opinion.

Reply to R2#3: *The choice of a composite primary outcome (as in Riello et al., 2020) and of an analysis aimed at the endpoint outcome is based on 1) the availability of prevalence data for the composite outcome in this population (Riello et al. 2020), 2) the nature of our randomization strategy and 3) organizational considerations. Indeed, with a randomisation stratified by recruiting centre we should obtain substantial similarity between the intervention group and the control activity group in terms of baseline assessment (as recruiting centres may differ for their history of COVID-19 contagions and for their response to the emergency, all of which in turn could affect workers' levels of distress). In addition, NCH look at efficacy in terms of proportion of workers in distress and potentially needing psychological assistance, rather than individual rates of improvement. Because we have opted to negotiate with NCH access to all of our potential participants, we have formulated our main target in a way that could be most informative and useful to the organisation. For these reasons we prefer to keep our primary outcome and analysis strategy as they are. However, we also fully understand and share this Reviewer's interest in effects at the individual level. We have now included in the protocol a secondary analysis of each scale which includes controlling for its baseline value and assessing the global statistical significance of treatment through seemingly unrelated regression (Zellner, 1962) followed, in case of statistical significance, by regression on each single scale (please see section 2.7.3).*

R2#4: The authors do not include details of the secondary analysis plans and they justify this decision: "As the secondary outcomes are included for exploratory purposes, only the hypothesis and analysis plan for the main outcome will be preregistered with the protocol, in compliance with the Registered Report format requirements." However, it is conventional for these to be prespecified and certainly to appear in the methods section of a paper. Personally I think it would be better to include details on how these will be analysed at this stage but this can be decided by the study team/editor.

Reply to R2#4: *We have now included in the methods section the plan for our secondary analysis, as in conventional trial reporting (please see section 2.7.3).*

R2#5: I struggled to follow the recruitment process and how the required sample size will be achieved. Will all workers in participating NCH be contacted at once, and all of those that wish to be will be assessed for eligibility? If many more than 164 are eligible, will all still be included in the trial? Since the primary analysis is (effectively) intention to treat but will exclude some participants after randomisation, I think the target sample size needs to be adjusted up to account for potential drop out due to participants never accessing the intervention.

There is also the potential for loss to follow-up after randomisation and after accessing the intervention. How will those participants be treated in analysis? Some loss to follow-up should be assumed and adjusted for in the sample size target if appropriate. Treatment of missing outcome data in general needs to be described.

Reply to R2#5: *We have now clarified in our manuscript (please see section 2.1.3) that the recruitment of individual participants (workers) will happen at once via participating NCH soon after receiving IPA. Because a) we are aiming for a minimum sample size of 178 (not including attrition and non-compliance) for the preregistered analysis of the primary outcome and b) some benefit can be expected from both the intervention and the active control activities, we will over-recruit. As now specified in the power calculation section (please see section 2.1.2), our minimum sample size for recruitment (i.e. taking into account the expected attrition/non-compliance rates) is set at 210, but if more workers than 210 are eligible and wish to participate, they will be included in the trial. We have now also clarified that, as in Riello et al. (2020), there will not be missing data at the item level but there could be data missing at the outcome measure level. Any participants missing primary outcome data (i.e. for one or both measures) will be excluded from our main preregistered analysis and counted as dropouts (i.e. they will contribute to attrition rates) (please see section 2.7.1). In case data were available for only one of the scales contributing to the primary outcome, such participants will be excluded from the main preregistered analysis but will be included in the secondary analysis concerning their available data. In case data were available for only one or two of the secondary outcome scales (our protocol includes 3 secondary outcome scales in addition to the 2 contributing to the primary outcome), participants will be included in the analyses concerning their available data (please see section 2.7.3). In order to access the primary outcome scales and the secondary outcome scales, participants will have to complete the demographic questionnaire, including questions about expectations, assistant helpfulness and perceived efficacy. Therefore, such data will be available for all participants included in our analyses.*

R2#6: The facilitator represents a potential source of bias in the study, in that they are aware of the intervention assignment and interact with the participants: as such, they could treat participants in different groups differently (and could plausibly favour the intervention). This should be further discussed and mitigated if possible. Would it not be possible to simply send all participants a message/email reminder once per week, rather than interacting with them more substantially? The fact the usefulness of the facilitator will be rated may provide a check on whether or not the facilitator treated the groups equally (this could be added as an additional validity check).

Regardless, the role of the facilitator needs to be explained more fully. The text states “The alternative activity is designed to 1) partial out improvements that could be elicited by facilitator attention, individual engagement and expectations rather than by specific content”. Is the facilitator not part of the SH+ intervention? If they are, then it’s not clear why the effect of the facilitator needs to be differentiated from the audio and reading material. If the facilitator is not part of the intervention, then why is a facilitator included at all (as when the intervention is used in a real world setting there would not be one)? Details of the facilitator’s participation and training should also be clarified. What will the facilitator message the participants? What will the facilitator say on the call in the case of no response? What training will the facilitator be given, and can this training material be shared so that the experiment can be repeated and so that the intervention can be used in practice in the same way it was tested if it shown to work?

Reply to R2#6: *The Reviewer is absolutely correct in pointing out that the facilitator, as described in our previous manuscript, may be a source of bias. Based on the Reviewer's suggestions, we have now downsized such role to one that requires no specialised training and renamed her as "assistant". As mentioned in the revised version of the manuscript (please see section 2.5), the assistant will be masked to the intervention assignment and will not speak directly to participants. Her role will be limited to sending message/email reminders to all participants (equally to the intervention and to the control activity group). The log of these messages will be recorded and the text of the messages will be made available for replication. In previous SH+ studies a facilitator was needed and played a crucial role due to the delivery modality (i.e. in groups) of the intervention. Our assistant will be purely functional to helping participants in both the intervention and the control activity group maintain their focus and commitment - work shifts and pre-existing daily commitments notwithstanding - and will be an easy-to-replicate feature in a potential larger-scale deployment of the intervention.*

R2#7: I would like to know more about the risk of contamination of the intervention amongst participants within study centres. Is it possible that participants will discuss the intervention amongst themselves and become unblinded to the study arms? Perhaps estimating the proportion of staff in each centre that you are expecting to recruit to the study would be useful in assessing this.

Reply to R2#7: *There can be two effects of contamination: one is related to unmasking, which we exclude in this trial, since all interventions are presented as active and will require some reflection on relevant issues/emotions (i.e. it will not be obvious to our participants, who do not have a research background, that one is actually a control activity). The other is subtler and related to workers sharing the contents of their assigned activities with other workers who belong to a different arm. To decrease the probability of this happening, we have now added a note to the informed consent form in which we expressly ask to avoid sharing any information and contents with colleagues for the duration of the trial (please see files in the Outcome measures folder on the OSF). We should also mention that, in our experience, healthcare workers tend not to share information about their mental health with one another, to avoid stigma in their work environment as, in Italy, mental health issues are often considered taboo or seen as of secondary importance. This may be also true of other countries. A study by Chen and collaborators (2020) reports that, despite the high prevalence of psychological disorders found in China at the beginning of the pandemic, healthcare workers were reluctant to participate in group or individual psychological interventions that were being offered to them. Also, many nurses who showed signs of psychological distress maintained that they did not have any problems (Chen et al, 2020).*

R2#8: Before commencing the trial, the authors should register their study on a clinical trial registry. That they will do this and where and when should be stated in the protocol. I have included a CONSORT checklist with the methods items completed according to my assessment of the article and have noted items where I've requested more details (because the authors stated they would do the study according to CONSORT). It would be helpful if the authors could submit an updated checklist with their next submission and denote the page numbers where each item is reported (or an explanation of why it is not applicable). Obviously this is at the discretion of the editor and team but would certainly be useful. They may also wish to consult the SPIRIT statement for RCT protocols.

Reply to R2#8: *Thank you for filling and providing us with the CONSORT checklist. However, since several of the requested data are not yet available at this stage, we have now eliminated the reference to CONSORT from our Stage 1 manuscript; also, we will not proceed to register our study on a clinical trial registry (please see Associate Editor's point AE#2).*

Specific comments

R2#9: Title: the title should state that it is an RCT as per CONSORT.

Reply to R2#9: *Thank you, the title now states this is an RCT.*

Introduction

R2#10: Pg3 line 36: the acronym ACT should be defined.

Reply to R2#10: *This has now been defined a few lines before.*

R2#11: Pg3 lines 44-52 discuss another trial of SH+ in a different setting and population. It would be useful to summarise the results of the trial in more detail: for example the magnitude of the effects found and the specific endpoints used. The primary endpoint used in that study (K6 score) differs from that used in this study – the reasoning for using a different endpoint could be described. That trial used considerably more participants than the present study and it may be worth commenting on why in the current trial the sample size is much smaller. It would be useful to note whether the RCT mentioned on line 44 is the only RCT of SH+ currently available and, if there are others, to describe them. Clinical trial registries (e.g. clinicaltrials.gov or ICTRP) could be briefly searched to determine whether there are any other ongoing studies of the same intervention.

Reply to R2#11: *We added more details on the study by Tol and colleagues as follows:*

“As planned in the present trial, the assessments of the trial in Uganda were conducted at baseline, immediate post-intervention, and after three months. Findings from the trial in Uganda are promising, highlighting a significantly greater reduction in psychological distress measured with the Kessler-6 (K6) rating scale after intervention (β -3.25, 95% Confidence Interval (CI) -4.31 to -2.19; $p < 0.0001$; d -0.72) and after three months relative to the enhanced usual care (β -1.20, 95% CI -2.33 to -0.08; $p = 0.04$; d -0.26). In addition to the decrease in distress, the authors identified a significant improvement of PTSD symptoms at post-intervention (β -3.53, 95% CI -4.67 to -2.38; $p < 0.0001$; d -0.68) and at three months (β -1.55, 95% CI -2.87 to -0.24; $p = 0.02$; d -0.30). [16]. That said, these findings need to be replicated in our target population group as they were produced in a different population group (refugees) and in a different setting (refugee camp in a low-income country [16]). Even though there are some commonalities in the choice of rating scales for measuring trial outcomes (i.e., WHO-5), the population group of the Ugandan trial (mainly composed of severely distressed women), together with cultural peculiarities and limited amount of resources led to the choice of different outcomes and different instruments for measuring outcomes.”

As the team in Verona, being a WHO Collaborating Centre, routinely interact with WHO Headquarters in Geneva, we can confirm that the trial of Tol et al. in the Lancet Global Health is the first trial testing SH+. Two other trials coordinated by Corrado Barbui have just been completed and are currently submitted to scientific journals (Purgato et al., 2019). To date, there are no other trials available in the scientific literature on this intervention.

R2#12: Pg3 line 53 to next page describe the intervention and control arm. The rationale for using an active comparator is given, but the specific control chosen is not well justified: the authors are creating an intervention, but is there no established method with demonstrated or possible effect that SH+ could be compared against? If there are no established methods, then a preferred control arm might more closely approximate usual care (e.g. in Tol et al 2020, ref 16 they use ‘enhanced usual care’) and in registered studies found on clinicaltrials.gov (NCT03571347 and NCT03587896) use enhanced usual care. Since nothing is known about the efficacy of the proposed control arm, a finding of no difference between the groups would be difficult to interpret – both arms could be effective, or both ineffective, and as such no useful finding generated. I do appreciate that changing this may not be possible due to the requests from the NCH etc.

Reply to R2#12: *Materials for the control group have been created ad hoc to closely match the materials of the intervention (please see the sample provided in the Sample Materials folder on the OSF), the control group will access the materials by the same interface as the intervention group, receive the same assistance as the intervention group and be required the same time commitment as the intervention group. From an experimental perspective, this is more informative than comparing the intervention group to another intervention with different materials and requirements. Our control activity is active in the sense that it engages participants but we do not expect its effects to be systematically related to its content, as it is not a professional intervention targeted to provide stress-coping tools. It is however well-matched to the*

superficial (and potentially active, in a non-specific way) aspects of SH+, and this could only be achieved by creating an ad hoc control intervention. By comparing SH+ with any other non-matched active intervention, available from the literature, the risk of finding no difference between the groups would be higher, and the interpretation of results more difficult (e.g. any significant difference could be due either to non-controlled superficial characteristics or to the actual contents of the intervention). With regard to the comparison between enhanced usual care and the intervention whose efficacy is being tested, we believe a trial of that kind aims to answer a different question than the one we are posing. In other words, it does not test the efficacy of SH+ contents per se but of SH+ as a whole against doing (almost) nothing (which makes a lot of sense in a humanitarian emergency setting, where the scope of possible interventions is more limited). However, it cannot distinguish between the efficacy of a generic involvement in any kind of activity, of some superficial characteristics of SH+ (e.g. time spent away from daily preoccupations, a booklet with nice figures, soothing voiceover, etc.) and content-specific efficacy of the SH+ (i.e. ACT-based reflections and activities).

R2#13: Pg 4 lines 52-54: “as a psychological intervention to increase the ability of nursing and care home workers to cope with stressful situations during the second wave of COVID-19 contagions”. It may be that we are no longer in the second wave of the pandemic by the time the trial is completed and the intervention can be used, so using a less restrictive term could be better (e.g. “during the pandemic”).

Reply to R2#13: *Done, thank you.*

Materials and methods

R2#14: Pg5 lines 38 – 44: this information is background info and is not relevant to the sample size calculation (the subheading it is under).

Reply to R2#14: *The lines have now been eliminated.*

R2#15: Pg5 lines 57-60: “Also, whatever the current prevalence, the SH+ intervention will provide participants with very simple but effective tools to cope with an acute or chronic stressful situation at work.” We do not yet know if the SH+ intervention will be effective. Also, prevalence in each group influences the required sample size. This statement should be removed in my opinion.

Reply to R2#15: *Removed.*

R2#16: Pg6 lines 8 – 12: Why is a relative risk reduction of 25% expected? How does this compare to effect sizes of other studies of psychological distress in general, and specifically to the other SH+ trial mentioned in the introduction.

Reply to R2#16: *The trial conducted in Uganda identified a strong and significant effect for the primary outcome immediately after the end of the intervention (β -3.25, 95% Confidence Interval (CI) -4.31 to -2.19; $p < 0.0001$; d -0.72) that was smaller but still significant after three months (β -1.20, 95% CI -2.33 to -0.08; $p = 0.04$; d -0.26). We added details on this trial in the manuscript, according to R2#11. Continuous data, however, might be difficult to interpret by a clinical audience who should then implement the results of our research into evidence-based practice. For this reason, and to ensure the highest applicability of this research into practice, we considered studies using dichotomous outcomes. Please, see also our reply to SSE#4.*

R2#17: Pg6 lines 13 – 14: power of 80% is conventional and acceptable (so does not have to be revised). However, since the intervention is probably quite cheap to deliver, I did wonder if the authors had considered using a higher power (e.g. 0.9 which would require ~106 participants per groups) as 20% is quite a high false negative rate. Below I note that a onesided

test may be appropriate which would increase power for this sample size if you choose to do it.

Reply to R2#17: *We agree with the Reviewer's suggestion and have modified the protocol by making the test one-sided and increasing the power to 0.9.*

R2#18: Pg6 lines 16 – 19: “Since we cannot provide an accurate prediction of the attrition/noncompliance rate with this population (in our previous epidemiological study, the rate was 5% but the current study includes two crucial data locks for the preregistered analysis)”: which study is “our previous epidemiological study” referring to? It should be cited here. If it is Riello et al 2020, I do not see how any information in that study is relevant to noncompliance (since there was no intervention) so the sentence should be reworded.

Reply to R2#18: *Given that our preregistered analysis is effectively an “intention to treat” one, attrition will be measured on the basis of responding rates to primary outcome measures. The primary outcome is measured with the same questionnaires as those included in Riello et al.’s (2020) study. Besides the intervention in-between (which will formally contribute to the attrition rate only minimally, due to some participants deciding not to proceed after providing consent but before viewing any of the materials), this study is otherwise very similar to that of Riello et al. (2020). Unlike in Riello et al. (2020), however, participants will have to fill twice (rather just once), and 7 weeks apart, the primary outcome questionnaires in order to be included in our preregistered analysis. A 5% dropout rate in Riello et al (2020) could thus hint to an expected attrition rate of at least 10%. Other considerations should then be added to a balanced (yet still very hypothetical) estimate, such as the attrition rates found in fairly similar RCTs, but also the expected strength of motivation in our prospective volunteers, which belong to a population that has never been studied in an RTC of this kind before. Based on all these considerations, we settled on an expected 15% attrition (including non-compliance) rate (please see section 2.1.2).*

R2#19: Pg 6: “Randomization will be stratified by centre and the RCT will be conducted in accordance with the Consolidated Standards of Reporting Trials statement” – CONSORT is a reporting guideline, not a guideline for conduct of RCTs. I would change the terminology to be “reported according to CONSORT” (and then ensure that it is reported according to consort). This statement is not relevant to recruitment and inclusion criteria (the section it’s in) so may be better placed at e.g. the beginning of the methods.

Reply to R2#19: *This statement has now been eliminated.*

R2#20: Pg6 lines 30-35: in recruitment and RCT inclusion criteria, the centres where recruitment will and has taken place should be listed and the prevalence study should be referenced when mentioned.

Reply to R2#20: *We have now referenced the prevalence study but we will not publish the list of the centres where recruitment has taken place to respect their privacy (see also Riello et al., 2020).*

R2#21: Pg6 lines 46-55: are there any exclusion criteria? In the other SH+ trial (Tol et al 2020) they state that “Self-Help Plus is not intended for people with complex mental health problems (such as psychosis) or those at imminent risk of suicide.”. Should this be addressed in the present study?

Reply to R2#21: *The Reviewer is correct: we do not exclude patients with a psychiatric condition because the treatment has no contra-indications (Epping-Jordan et al. 2016) and thus everyone can participate in it. However, we have now added a note to the pre-enrolment information form to clarify that this intervention does not replace the need for psychiatric care (please see documents in the folder Outcome measures on the OSF), in order to inform potential participants with a psychiatric diagnosis that they will need other treatments and cannot rely on this alone.*

Furthermore, at the end of the study, the participant will have the possibility to contact us for feedback on the questionnaires, or for any type of problem. If any of the participants presents serious signs of psychological distress, they will be contacted by our psychologist who will provide further guidance and advice.

R2#22: Pg7 line 7: “To avoid contamination, only one person per household will be randomised.”

How will the study staff know whether someone from the same household has already been randomised? I could not see a question in the OSF material about this.

Reply to R2#22: *We were originally thinking of gathering this information from the recruiting centres through their address database, however, on second thought, it would be best if we obtained the information from the participants themselves (as also colleagues from different care homes participating in our study may be living in the same household).*

We have now added the following request to the survey:

“Do you live with colleagues from the same or a different care home?”

In order to enable us to correctly interpret the results of this study, we kindly ask you to provide their name(s) here. Please be assured that this information will be kept strictly confidential and used exclusively for research purposes.”

R2#23: Pg7 line 13: “the randomization list will be accessible only to the data manager and to the website manager, for the assignment of activities to individual participants”. Which of the website manager and data manager will assign participants to interventions? Both? Why?

Reply to R2#23: *We have now amended this aspect. The website manager is the only person who needs to know what is assigned to whom (as he has to provide individual access to the correct set of materials by associating access credentials to email addresses). Therefore, while the data manager will pass on participants’ contacts to the website manager, the website manager will perform the randomisation and assign participants to interventions. Thus, anybody else who has contact with the participants or handles the data (recruiter, assistant, data manager and data analyst) will be blind to the randomization outcome.*

R2#24: Pg7 lines 5-23: Will the individuals making assigning participants have any access to information about baseline assessment? If not, could you please describe how they will be unaware of this information (e.g. they will simply receive the email addresses of eligible participants, randomly ordered).

Reply to R2#24: *The website manager will receive from the data manager a list with the email addresses of eligible participants, divided for recruiting centre, and will perform a stratified randomisation with Stata. He will not have access to baseline assessment outcomes (collected via separate web software), which will be handled by the data manager and passed on to the data analyst (please see section 2.1.4).*

R2#25: Pg7, lines 51:56 and Figure: this information is not specific to the characteristics of the intervention (the subheading). I would suggest it is moved to the start of the methods where it would provide a nice overview of the planned approach.

Reply to R2#25: *Done, thank you.*

R2#26: Pg8 line 11 states that the activities will be disclosed to participants on week 3 of the RCT. From the figure I would have thought they’d be disclosed on week 2, when the intervention begins. Please clarify.

Reply to R2#26: *The Reviewer is correct, it is Week 2. We have now amended the typo.*

R2#27: Pg8 SH+ description. For clarity, I would suggest that this section be dedicated to describing the actual intervention that will be used in the study, rather than background information on it which could be included in the introduction (i.e. lines 29-49 could be elsewhere as they don’t describe the intervention given to the participants).

Reply to R2#27: *Done.*

R2#28: A sample of materials is provided, including very helpfully in English. However, the exact method of delivery and full details of the intervention and control are not provided. A full description and preferably materials is essential to allow others to reproduce the study or to

use the intervention in the event that SH+ is found to be efficacious. As I understand it, participants will access the information through a website where they need to log in. Can this website and the intervention in its usable form be shared? If not, it should all be described in more detail. Other details like how long the intervention takes and the frequency with which the participants will be expected to engage with it should be given. This comment applies both to the intervention and control arm.

Reply to R2#28: *We have already provided part of the materials (outcome measures) and intervention samples for this review process but, as stated at the end of our Stage 1 manuscript, all materials and/or necessary details for replication will be made available on the OSF at the end of the study. At the moment, we cannot say whether we will be able to cover the maintenance costs of our dedicated website beyond this pro bono study, however we would certainly consider sharing and maintaining the intervention in a ready-to-use form for a larger roll-out in case of a positive result.*

R2#29: Page 11- 13: there are a lot of different scales that participants are expected to complete. How much time are these likely to add to assessments for participants? Have the authors considered that this might put off some participants and reduce completion rates? Notwithstanding the point about overburdening participants, could it be worth collecting information on reasons for dropout/lack of engagement with the intervention? This seems potentially more valuable to me than the different measures of well-being and anxiety.

Reply to R2#29: *The baseline assessment will last about 20 min, the following will be slightly shorter, due to the elimination of some demographic questions that do not need to be repeated. This seems a reasonable time for participants who are motivated enough to commit to an intervention that requires 1-2 hours per week, for 5 weeks (a rather demanding task for this population, at this point in time). Participants are given the possibility to leave comments at the end of the survey and some extra questions have now been added regarding expectations, engagement and perceived efficacy (please see Reply to SSE#1).*

R2#30: Page 12 table 1: the timing of the post-intervention assessment (column 3) should be clarified. Based on Figure 1 I think it means 1 week.

Reply to R2#30: *Yes the Reviewer is correct, it means 1 week. We have now changed the header of the Table.*

R2#31: Page 13 validity checks: it might be useful to list the validity checks and the specific criteria that will be considered as 'valid' vs. not so that it will be clear in the results whether the checks were passed. You could consider adding something about level of engagement in the materials – for example, if all the participants in the SH+ arm log in once to the materials but do not use them after that, this might indicate there is some issue with the delivery of that specific intervention via its website. You could also ask someone to test the materials (i.e. access and quickly go through them) in the format they will be used by participants to check for obvious software issues. You could also collect data on what participants liked and didn't like about the intervention to inform future interventions and give an opportunity to highlight major issues with its delivery.

Reply to R2#31: *We believe that the criteria for inclusion in the preregistered analysis are fairly clear, and we have added some specification in section 2.4 concerning a check to prevent contamination from household sharing. In our result section, the Reviewer will find a complete and detailed list of the checks applied. Thank you for the advice on pre-testing materials and procedures before rolling-out the intervention: we fully agree and confirm that this is an approach we routinely adopt with any studies in our labs.*

R2#32: Page 13 line 32-35: if more than one response is sent from one device, what will be done?

Reply to R2#32: *The response collection software does not allow the sending of more than one response from one device.*

R2#33: Page 14, table 2: why is the analysis two-tailed? The hypothesis is framed as a reduction in proportion in the intervention group (rather than a difference in either direction), which

seems sensible to me as it seems unlikely that an increase in proportion in the intervention group vs control would be observed. Using a one tailed test would give more power (for the same sample size) and would I think test the stated hypothesis. Note that if this is changed the power calculation needs to be updated.

Reply to R2#33: *We agree with the Reviewer's suggestion and have modified the Protocol by making the test one-sided and increasing the power to 0.9.*

R2#34: Page 15, lines 41 - 44: generally, it is not a good idea to rely solely on the p-value for the primary outcome. Can the authors include a measure of precision of the estimated effect size (e.g. 95% CI) or explain why this is not appropriate? This will be needed as part of the CONSORT statement when reporting results and it is best to prespecify it

Reply to R2#34: *We have added the condition of the upper limit of the CI being within 25% of the estimated value of risk ratio with a probability of 90%.*

References

Chen Q et al. 2020 Mental health care for medical staff in China during the COVID-19 outbreak [published correction appears in *Lancet Psychiatry* 7, e27. *Lancet Psychiatry* 7, e15–e16. (doi:10.1016/S2215-0366(20)30078-X)

Epping-Jordan JE, Harris R, Brown FL, Carswell K, Foley C, García-Moreno C, Kogan C, van Ommeren M. 2016 Self-Help Plus (SH+): a new WHO stress management package. *World Psychiatry* **15**(3), 295-296. (doi: 10.1002/wps.20355)

Purgato M, Carswell K, Acarturk C, Au T, Akbai S, Anttila M, Baumgartner J, Bailey D, Biondi M, Bird M, Churchill R, Eskici S, Hansen LJ, Heron P, Ilkkursun Z, Kilian R, Koesters M, Lantta T, Nosè M, Ostuzzi G, Papola D, Popa M, Sijbrandij M, Tarsitani L, Tedeschi F, Turrini G, Uygun E, Välimäki MA, Wancata J, White R, Zanini E, Cuijpers P, Barbui C, Van Ommeren M. 2019 Effectiveness and cost-effectiveness of Self-Help Plus (SH+) for preventing mental disorders in refugees and asylum seekers in Europe and Turkey: study protocols for two randomised controlled trials. *BMJ Open*. May 14; **9**(5):e030259. doi: 10.1136/bmjopen-2019-030259.

Riello M, Purgato M, Bove C, MacTaggart D, Rusconi E 2020 Prevalence of post-traumatic symptomatology and anxiety among residential nursing and care home workers following the first COVID-19 outbreak in Northern Italy. RSOS <https://doi.org/10.1098/rsos.200880>

REVIEWER 3 (R3)

The manuscript "Effectiveness of Self-Help Plus (SH+) in reducing anxiety and posttraumatic symptomatology among care home workers during the COVID-19 pandemic" (RSOS-210219) proposes the evaluation of a WHO-developed intervention package. The manuscript generally reads well and the major points to pin-point the main analysis have been described. The evaluation of the efficacy of a programme in a new target population is a valid research question.

R3#1: There are some aspects I comment on in more detail below, mainly for documentation of the evaluated aspects, but I want to highlight that the key questions I have are (i) why an surprisingly simple trial analysis was chosen, (ii) why contamination at centre-/workplace-level is accepted, and (iii) why the primary outcome is evaluated at such a short time frame (1-week).

Reply to R3#1: *Thank you for highlighting these issues, we have addressed them all in our replies below.*

1) Primary outcome

1a) The power calculation was performed in proprietary software and could not be replicated. Based on the OSF-provided code the calculation is correct. The complete syntax for the planned test, also in line with the description in chapter 2.7.2 (making 5% sig level and the planned chi2 test explicit), would have been: `power twoproportions 0.8 0.6, power(0.8) alpha(0.05) test(chi2)`

Using other calculators with these specifications results in the described target sample size.

R3#2: 1b) The team should mention that the test is two-sided (often forgotten when dealing with chi2 distributions).

Reply to R3#2: *This has now been changed to a one-sided test following R#2's suggestion.*

R3#3: 1c) The section in 2.7.2 should entail a statement about how the result of the test will be interpreted. As the test is two sided, it is not pre-defined which constellation of proportions and significance test result will be interpreted as in line with the hypothesised effect of the intervention.

Reply to R3#3: *The test is now one-sided and a statement about how its result will be interpreted has been added to section 2.7.2 accordingly.*

R3#4: 1d) While the logic of the testing strategy is sound, it rests on a number of assumptions and decisions that have already been made. The dichotomisation of the outcome is usually considered a severe limitation in the trials literature and should be avoided where possible (as it also decreases power). The team is also not controlling for anxiety and posttraumatic stress symptoms at the start of the study, which is counter-intuitive given the detailed introduction provided around the development of stress levels in the target population. This latter point is again important as it again reduces statistical power and it will make the interpretation of the results more difficult, as groups can obviously start at different proportions even with randomisation present, i.e. the endpoint alone is not on its own relevant for the decision on the effectiveness of the programme (important for 1c as well). I want to stress: within the logic of the study this is fine, but a stronger testing strategy could have been developed.

Reply to R3#4: *We have chosen a binary outcome on the grounds of clinical reasons (please see our Reply to SSE#3), and we have not controlled for baseline values in our main analysis given that differences at baseline are due to randomization.*

Nevertheless, we agree with the Reviewer on possible imbalances due to the effect of chance. To account for this, and for the continuous nature of our variables of interest, we decided to include, among secondary

analyses, regressions to estimate the effect of SH+ on GAD-7 and IES-R controlling for their values at baseline.

R3#5: 1e) With 1-week, the evaluation of the effectiveness is on the extreme short-term side for psychosocial workplace interventions, in particular self-guided ones. More justification for this choice could have been provided.

Reply to R3#5: *We have now clarified that since this is a relatively light, short-lasting intervention, we expect its effect to peak immediately post intervention (though we do include a test for its medium-term efficacy). Given the agility of the intervention it could be repeated as necessary in other emergency situations, if beneficial effects are found in the short term (please see section 1.1).*

R3#6: 1f) Using a composite measure (and a dichotomous one, see 1d) is usually fraught with difficulties. But due to its logical definition as "either type of symptom" the typical problems of one dimension changing but not the other etc are avoided.

Reply to R3#6: *Please see our Reply to SSE#3 on the use of a composite and dichotomous measure. Thank you for noticing that its logical definition overcomes typical pitfalls.*

2) Experimental design

R3#7: 2a) If I understood 2.1.4 correctly, the study does not address contamination sufficiently. While contamination at household level is avoided (2.1.4, line 2), contamination at centre level (healthcare workers talking to each other, sharing materials etc) is not addressed at all. This seems to me to be a severe shortcoming of the design.

Reply to R3#7: *Please see our Reply to R2#7.*

R3#8: 2b) I may have missed it, but I did not find a description of how the data will exactly be collected (which online system, who has access at which point etc).

Reply to R3#8: *This information has now been added to the manuscript (please see section 2.5).*

R3#9: 2c) I don't think it is stated how many potential participants will be approached or whether this study will approach as many as needed to get to the required number of responses (as per power analysis).

Reply to R3#9: *We have now updated the relevant section and also provided a recruitment target, based on our best assessment of attrition and non-compliance risk in this population (please see section 2.1.2).*

3) Intervention and control

R3#10: I checked the report of the intervention and control conditions against the TIDieR checklist. The only thing missing were the materials for the control group (the link provided led only to intervention group materials).

Reply to R3#10: *A sample of the control group materials was actually provided but since they look and sound very similar to those of the intervention group, the Reviewer might have mistaken them for the intervention. In case, instead, the Reviewer was looking for an ad hoc English translation of the provided control materials, we have now created and added them to the folder on the OSF, specifically for the Reviewer.*

MINOR:

R3#11: I am not sure what the team means with the "two data locks". Do the authors mean that the data were logged at those time points?

Reply to R3#11: *We mean “at the 1-week follow-up and 3-month follow-up”). This has now been clarified in the text.*

R3#12: To leverage the power of the pre-registration, the planned analysis for the secondary outcomes could be described as well. It sounds like the authors have already settled on a plan for action, i.e. it could easily be added.

Reply to R3#12: *Done.*

R3#13: page 13: "A facilitator will check in regularly with the participants via WhatsApp, text messages and phone calls, to monitor their progress and to encourage compliance" Will this be done in both groups?

Reply to R3#13: *Yes, in both groups. We have now downsized the role of the facilitator (now called “assistant”) and eliminated the phone calls from her tasks.*

Rovereto, 19th October 2021

Dear Professor Chambers,

Please find enclosed our Stage 2 manuscript, in which we have extended the Stage 1 manuscript by adding a Results and a Discussion section. All the other sections are identical to the Stage 1 accepted manuscript except for the following:

1. Extension of the abstract, to include the main result and message. The central result (i.e. the result of the preregistered analysis) has been flagged and clearly separated from the results of the exploratory analyses. Our conclusive considerations build on the result of the preregistered analysis.
2. Changes in verb tenses. In particular, future tenses have been replaced with past tenses and/or conditional (the former to indicate what has been done, e.g. p.2 “it was tested primarily”, the latter to indicate what actions would have been undertaken, if the circumstances had required so, e.g. p.14 “exclusion would therefore be performed”).
3. Corrections of typographical or other types of minor errors – with no bearing on the content (e.g. p.16 “Internal Review Board” instead of “Ethics Committee”, as this is technically more accurate, although the term Ethics Committee – which deals more specifically with research on clinical populations and operates at regional level in the national healthcare system - is widely used to indicate University or other institutions IRBs).
4. Update of time-sensitive information (e.g. p.3 “especially in the poorest parts of the World” and “31st December”).

5. Harmonization of terminology throughout the paper (e.g. p.3 “effectiveness”, consistent with the title, rather than “efficacy”).
6. Simplifications of convoluted sentences, to improve clarity without changing the meaning (see e.g. p. 14 “We would exclude participants based on careless responding...”) or the transposition of sentences from one place to another, to improve clarity (e.g. p.3 “The SH+ was originally...”).
7. Inclusion of the start and end dates of the RCT both in Figure 1 and in the main text (p.6) and specification of the overall duration of the trial (p.6). Two extra weeks have been added to the overall trial duration as a result of a delay to the start of post-test 2 (which therefore took place 3.5 months, or 14 weeks post intervention). Please note that this could not have changed in any way the result of our preregistered analysis, which was focused on the primary outcome at 1-week post intervention.
8. Specification needed following the necessary addition of an extra response option to questions 15-19 (“I cannot answer”); see p. 18 “except for a few who chose not to give a rating”.
9. Several extra details on the exploratory analyses, that are necessary in order to better understand the Result section. Please note that, compared to the approved protocol, we have added only Cochran’s Q test and the related box plot to the exploratory analyses. This decision was taken after seeing the data and it has been flagged both in the Methods and in the Results section.

Any changes to the previously approved text have been highlighted in the manuscript for your and the Reviewers’ convenience, and associated with a note when necessary. We have uploaded the RCT data file and the scripts used to perform the reported statistical

UNIVERSITY
OF TRENTO - Italy

Department of Psychology
and Cognitive Science

analyses at the following OSF link:

https://osf.io/gdpsw/?view_only=b289c193ac494b0dbc1c78dec78c8a7a. All the materials used to collect the data and the outcome measures have also been uploaded and are available at the same link. Yellow highlights in the files with the outcome measures indicate necessary clarifications that were missed out in the previous version. Only in the case of questions 15-19, there has been a modification to the protocol, with the addition of an extra response option (“I cannot answer”), in order for the question to make sense also to potential responders who had withdrawn from the intervention.

We are very mindful of the urgency of publishing this work to inform the field on the one hand and to enable the timely planning of further intervention studies on the other. We hope that you and the Reviewers will find our Stage 2 manuscript to be suitable for publication in the Royal Society Open Science journal and we will be looking forward to hearing from you.

Thank you very much in advance for your time and your feedback,

Marianna Riello, Marianna Purgato, Chiara Bove, Federico Tedeschi, David MacTaggart,

Corrado Barbui & Elena Rusconi

Appendix E

Reply to Editorial Office:

EO#1: Please also be aware that we require all authors to have an active email address at the time of submission and acceptance, but note that chiara.bove@studenti.unitn.it is currently not receiving emails from the journal. Please can you either check that this email address is correct, ensure emails from the journal are 'white-listed' by the relevant IT providers, or supply an alternative email address at revision?

Reply to EO#1: Thank you for making us aware of this. After our Stage 1 submission, Chiara Bove's address has changed into: chiara.bove@alumni.unitn.it and has now been updated on the system.

Reply to Associate Editor:

AE#1: Two of the three reviewers from Stage 1 were available to evaluate the completed Stage 2 manuscript. As you will see, the comments are broadly very positive while noting several areas that would benefit from clarification, principally in the reporting and interpretation of the results (e.g. justification of certain exploratory analyses; ensuring that confirmatory and exploratory outcomes are clearly distinguished; ensuring that the interpretation of exploratory analyses is appropriately evidence-bound; consideration of limitations in the Discussion). All issues appear to be readily addressable through a round of careful revision. Provided the authors are able to respond comprehensively to all concerns, a final Stage 2 decision should be forthcoming without requiring further in-depth review.

Reply to AE#1: We are very grateful to the Editor for returning to us Reviews of such timeliness and quality. We have now addressed point by point all of the Reviewers' concerns, as detailed below, and we hope that both our arguments and changes will be found satisfactory.

Reply to Reviewers:

Reviewer: 1

This paper presents the second stage of a registered report on the effectiveness of a self-help intervention (SH +) versus an active control in reducing anxiety and PTSD symptoms in care home workers during the COVID-19 pandemic. From the pre-registered primary analysis, the authors did not find evidence that the SH + intervention was more effective than the active control. This study makes an important contribution to the literature in testing the effectiveness of the SH + intervention in a randomised design with an active comparison group. Based on their findings the authors speculate that previously observed positive effects of the SH + in studies without control groups may be due to non-specific factors.

The authors have made minor changes to the introduction section. For the most part these changes were warranted in updating the text to reflect the current situation. The stated hypotheses are the same as the approved stage 1 submission, and the data are able to test the proposed hypotheses. The authors adhered to the registered experimental procedures except for a 2-week delay to the planned timepoint for the time 2 follow-up. The authors have been transparent about this change and note that it does not influence their primary pre-registered outcome.

R1#1: I found the results section to be overly dense which made understanding the researcher's findings difficult. I think it would be beneficial to include some of the results in supplementary materials. For example, the authors could note that controlling for imbalance at baseline did not impact their findings in the main manuscript and include the details of these analyses in supplementary materials. It would also be informative to highlight which analyses were registered

versus exploratory in the results section.

Reply to R1#1: We agree with the Reviewer on the usefulness of a Supplementary file, where we have now moved the analysis on imbalance. All analyses have to be considered as exploratory apart from the one on the main outcome (presence of symptoms at the post-intervention assessment), that is the content of Section 3.2. Results related to the non-exploratory pre-registered analysis are also highlighted in bold in Table 4. All of the exploratory analyses were pre-planned and mentioned in the Stage 1 version of the manuscript, apart from Cochran's Q test, as specified in note 5 linked to the Methods section. To improve transparency, note 5 has now been linked also to the description of Cochran's Q test in the Results section.

R1#2: The tables and figures were somewhat difficult to read in their present format. It would be helpful to either separate figure 3 into two separate figures depending on the outcome (% of mild symptoms and difference in percentage) or just focus on one outcome rather than have two y axes. Adding an additional row or column to highlight the timepoints in table 4 (as the authors have already done in table 5) would be helpful. Using more descriptive labels for the timepoints in the tables and figures would also help clarify the author's findings (e.g., baseline, post-intervention and 14-week follow-up rather than T0, T1, T2).

Reply to R1#2: In Figure 3 we now focus exclusively on the percentage of people with at least mild symptoms in each intervention arm by time-point, and show a single y axis. Table 4 has now been re-organized in accordance with the Reviewer's suggestion. We have also harmonised the labels for time-points across tables and figures, as suggested by the Reviewer (thus "T0" has been replaced with "baseline", T1 with "post-intervention", T2 with "14-week follow-up").

R1#3: I have not previously come across the Cochran's omnibus Q test conducted by the authors and am therefore not able to comment on whether this unregistered exploratory analysis was statistically justified or methodologically sound. However, at present I feel that this test is not informative in the way results are presented. I found it confusing that the authors report significant findings for this analysis except for between T1 and T2 in the alternative group, but do not provide any further comment on this. The descriptive statistics seem to indicate that the whereas the SH+ group showed a decline from T1 to T2 in mild symptoms in at least 1 scale (74.12% to 64.71%; Table 4) the alternative group showed relatively stable levels (76.47% to 74.67%) which would seem to indicate some benefit of the SH+ in the long-term. It may be that I am misinterpreting these findings, but I believe further commentary would be helpful in helping readers to understand the practical significance of this analysis if it warrants inclusion as an exploratory test. Aside from this I feel that the author's conclusions are justified given the data.

Reply to R1#3: We acknowledge that the results from Cochran's Q test may seem at odds with the lack of statistical significance of the tests comparing the percentage of people with at least mild symptoms at T2 between the two groups. It has to be considered, however, that our main outcome was at T1, and thus our power analysis was based on expected results at that time point. We have now expanded a note (note 15) linked to our Discussion in order to highlight the need of further studies using a follow-up time similar to the one we used in our study for T2.

Reviewer: 3

I thank the editorial team for inviting me to review the Stage 2 version of this paper and I congratulate the team to finish their study so closely within the confines of their registration.

I can confirm that the Introduction, rationale and stated hypotheses are the same as the approved

Stage 1 submission. The report adheres to the registered experimental procedures and minimal deviations from the original approach are transparently described and in my view largely (see comments below) within the affordances of such a practice-based research project. The presented data are able to test the authors' proposed hypotheses and the exploratory analyses are clearly separated in the report. The authors' conclusions are largely justified given the data and reported results (comments on smaller aspects below).

I provide my comments below separated into Major and Minor points as well as a personal observation relating to the results – the latter clearly separated as it may invite what feels like undue speculation on the results and is not a critique based on comparisons with the IPA manuscript.

__MAJOR__

R3#1: 1) Page 19, " Finally, only one (the first to enrol) of any two or more responding workers from NCH who shared their accommodation with one another were entered into the randomization."

This is a difficult point that would require at least further discussion as a limitation. This detail of the procedure was not registered – in the Stage 1 manuscript it was only specified that:

p. 7: "Those meeting the inclusion criteria outlined below, completing an initial survey (demographic information and baseline assessment) and passing the validity check at the baseline assessment (see section 2.4) will then be randomly assigned to one of the two activity groups. Randomization will be stratified by centre and the RCT will be conducted in accordance with the Consolidated Standards of Reporting Trials statement."

And it was later specified, p. 8, that:

"To avoid contamination, only one person per household will be randomised."

As conducted and reported, in study planning this should have been an inclusion criterion i.e. stating that the first to enrol from shared accommodation will be randomised.

As conducted now, this is effectively a randomisation breach as the protocol/Stage 1 paper defined everyone as randomizable.

As far as I understand, the likely effect for this trial will be minimal as the authors report on page 19 "This led to exclusion of a further two (N = 2) respondents from randomization (see Figure 2)." Nevertheless, this should at least be noted.

Reply to 3#1: We respectfully disagree with the Reviewer on this point. Indeed, the Stage 1 protocol stated very clearly that only one person per household would be included in the randomization. Accordingly, we had included a question in our protocol to ascertain whether a person lived with any colleagues. Had we excluded all people sharing their accommodation with a participating colleague, that would have been a breach of the protocol. Had we previously stated we would select the individual at random, including the first to enrol would have been a breach. However, in the approved protocol we only stated we would have included only one person per household and that is exactly what we did. At Stage 1 we had not realised, and neither did the Reviewers, that the inclusion condition had remained underspecified in text, so the current clarification ("the first to enrol") simply fixes an oversight at Stage 1. This has now been clarified in the revised manuscript by adding a note to the main text.

R3#2a: 2) p. 27: The discussion of secondary results starting with "In exploratory analyses,..." seems to go a bit far when compared to the registration as well as to the results that are reported in the manuscript.

2a) "Since the SH+ has proved to be predominant for these characteristics, even in a conservative test like ours (albeit in secondary analyses), we can hypothesise that its use might be effective for skilled healthcare staff working in long-term healthcare facilities beyond emergency situations."

2a-i) It is unclear to me what "predominant" means in this case and how this relates to the strong interpretation offered.

2a-ii) It is unclear to me which test is described as 'conservative' here. From a statistical perspective any analysis this can refer to would seem to be pretty liberal as it was not pre-registered.

Reply to R3#2a: i) we have now replaced "predominant" with "superior" (here we are referring to secondary analyses, where SH+ shows superior scores for perceived effectiveness, expectation fulfilment and engagement) ii) the Reviewer is correct as the word "test" was misleading; it has now been replaced with "approach" (in reference to the use of an active control group).

We report here the modified sentence:

"Since the SH+ **was found to be superior** for these characteristics (albeit in secondary analyses), even in a conservative **approach** like ours **where the SH+ group was compared to an active control group**, it may be worth to test whether its use **could be useful** for skilled healthcare staff working in long-term healthcare facilities beyond emergency situations."

R3#2b: 2b) "Any further speculation on these findings would be unwarranted,..."

It is unclear to me why this small paragraph was included as it contains such 'further speculation'. Especially, the link to stepped care models seems to be a long shot as the present study was not conducted in such a context.

Reply to R3#2b: this paragraph has now been eliminated.

R3#3: 3) p. 28: " Moreover, we provided a large sample of workers in NCH with a practical strategy to independently manage distress, which was perceived - according to secondary analyses - as more effective and engaging than an alternative but similarly demanding activity. If this was confirmed, the strategy could be adapted for many other groups, which is particularly important given the potential long-term consequences of the COVID-19 pandemic."

This interpretation seems to assign undue relevance to the results of analyses that were not pre-planned as the primary outcome. This should either be dropped from the manuscript or the wording revised to reflect the exploratory nature of the results.

Reply to R3#3: We agree with the Reviewer and have toned down our statements here. The paragraph now reads: "Moreover, we provided a large sample of workers in NCH with a practical strategy to independently manage distress, which was perceived - according to secondary analyses - as more effective and engaging than an alternative but similarly demanding activity. **It could be interesting to investigate such differences in perception in a study that focuses on them and their expected benefits as the main outcome.**"

R3#4: 4) p. 28 " Exposure to chronic stressors and multiple traumatic experiences may generate long-term consequences even after the acute phase of the pandemic, especially for workers in NCH."

I am not sure why this sentence is placed in the "strengths" section.

If kept, there should be a reference provided – the presented empirical statement does not follow from this study.

Reply to R3#4: We would prefer to keep this sentence in the " strengths" section, given that **it** is linked to the clinical importance of providing psychological strategies to be independently applied in the long term. Therefore, we have now added appropriate references [74,75], as suggested.

R3#5: 5) Table 5 and its description are difficult to match to the description of the methods applied in 2.7.3.2. If I interpreted correctly that these are the results connected to that part of the methods section, the linkage in terminology could be increased (also: the SUR results which are indicated as the first methodological step seem not to be reported).

Reply to R3#5: Please see Reply to R3#8.

MINOR

R3#6: 6) What is the RSOS convention for decimal places as a number of different formats are used.

Reply to R3#6: As we could not find specific instructions regarding the number of decimals on the RSOS website, we have made our choices more consistent throughout the paper by always reporting 2 decimal digits. The only exceptions to this rule are risk ratios and numbers related to statistical significance (where we have kept three digits since these are necessary in order to always have at least three significant figures and to assess significance in case the p-value is around 0.05) and statistics indicating number of people (where we have approximated to the nearest integer for meaningfulness).

R3#7: 7) p. 22, " The Cochran's test turned out to be statistically significant..."

Suggestion: "was statistically significant"

Reply to R3#7: replaced, thank you.

R3#8: 8) p. 24: " Joint analyses on the GAD-7 and IES-R scores controlling for baseline values did not show any statistical significance (p-value 0.544 at T1 and 0.254 at T2) in the SUR equation model."

Partly a question to the editor: Is it admissible to report results without reporting full results (coefficients, SEs, and in case of SUR especially the residual correlations)?

Reply to R3#8: The SUR approach was only used as an omnibus test, thus we only reported its p-values. Results for the single outcomes were only reported in case of the significance of such tests. These results relate to single ordinary least squares (OLS) regressions, controlling for baseline values and, in case of significance of the White's test for homoscedasticity, using robust standard errors. We would like to bring to the Reviewer's attention that the SUR method exploits information on the correlation between errors across the different outcomes measured on the same sample, leading to lower standard errors than OLS, therefore here we opted for a more conservative approach (please see Haushofer and Shapiro, 2016, *The Quarterly Journal of Economics*, 131 (4), 1973-2042, for a similar approach in analysing RCT data). However, for completeness, we have now also reported in our Supplementary Materials the results of the SUR analysis, that are well in line with the results reported in the main paper.

R3#9: 9) p. 28: "...the blindness of participants and of the majority of the researchers involved..."

I suggest to reformulate, as neither participants and researchers were physically blind (I presume...)

Reply to R3#9: thank you, this has now been clarified: "...the fact that the participants and the majority of the researchers involved were blind to the intervention allocation, and that the website manager was blind to participants' baseline data".

R3#10: 10) Table 2 is not introduced/linked in the text.

Reply to R3#10: Thank you, this has now been referred to in section 1.2.

R3#11: 11) Table 4 contains drop-out ratios, which seem not to be commented on/explained in the text accompanying the table.

Reply to R3#11: We have now specified in the text that risk-ratios were included in the Tables for all binary outcomes at all time-points.

R3#12: 12) Page 18, "Number of log-ins was dic[h]othomized (no log-ins vs. at least one log-in),..."

Typo as indicated and it is not clear why the distribution of the raw data is not presented as well.

Reply to R3#12: We thank the Reviewer for noticing the typo and for the question. We realized the above-reported text was erroneous and have thus replaced "no log-ins vs. at least one log-in" with "only one log-in vs. multiple log-ins". The variable was dichotomized after inspecting the login distribution and noticing a particularly clear discontinuity between one and more logins, suggesting a qualitative difference between the two categories. Crucially, an imbalance between groups at this basic level would disclose a potential difference in first-sight attractiveness between interventions.

R3#13: 13) p. 26 " In exploratory analyses, perceived intervention effectiveness, match with expectations and engagement result higher at the post-intervention assessment after receiving the SH+ than after receiving the alternative intervention."

Sentence unclear to me. A comma after "with expectations" could make it clearer but I still think that "result higher" is then not correct.

Reply to R3#13: Thank you, to disambiguate this sentence we have now replaced "result" with "are".

R3#14: 14) p. 26 "These factors could be represented by the provision of structured and interactive activities, contents that stimulate reflection and expression of emotions, guided exercises and the stimulation of multiple senses through visual and auditory channels"

If possible a references/ references would be good as these are all empirical claims for causal connections.

Reply to R3#14: Thank you. We have now rephrased the point as follows, and provided a suitable reference: "These factors were similar to the so called "common factors" of psychotherapeutic interventions, such as the provision of structured and interactive activities, contents that stimulate reflection and expression of emotions, guided exercises and the stimulation of multiple senses through visual and auditory channels [76] (see section 2.2).

R3#15: 15) p. 28: " Further, despite the fact that the drop-out rates were very similar between groups, proving intervention feasibility, we were not able to identify and analyse reasons related to the dropouts."

Similar question as above: no results are reported to support this statement and I question therefore whether it is admissible.

Reply to R3#15: We reported data and statistical inference for drop-out rates in Table 4, and described them in Section 3.3.3.

R3#16: 16) Typo in reference [22] " pschological"

Reply to R3#16: thank you, this has now been fixed.

R3#17: 17) Reference 58 ([58] Baum, C., Schaffer, M. (2009) Implementing econometrics estimators with Mata. Stata Users Group, United Kingdom Stata Users Group Meetings 2009) is a reference for the particular routine used in the analysis; for the introduction of SUR models probably a different resource should be provided.

Reply to R3#17: We thank the referee for pointing this out. It was an oversight on our side as this should have been indicated as reference 57 (which had already been introduced), rather than 58. We have now amended this and also reworded in “we tested the effect of SH+ on GAD-7 and IES-R by performing seemingly unrelated regression [57]” as “we tested the effect of SH+ on GAD-7 and IES-R by performing seemingly unrelated regression (SUR) equations [57]”, and “The effect of SH+ on WHO-5, CD-RISC and PSS was tested by performing seemingly unrelated regression [58]” in “The effect of SH+ on WHO-5, CD-RISC and PSS was tested by performing SUR equations”.

R3#PO: __Personal observation__

I was surprised that the team does neither comment on the large (albeit anticipated) drop-out rate as well as the secular trend. I mention them both, as they may be interrelated issues.

Taking the observed data on their own, both groups are 'getting better' over time. One of the reasons for the lack of an effect could be that the assumption of a stable comparator group was violated as both groups changed substantially – and the added effect of the intervention is not strong enough. But relatedly, this trend offers the potential for systematic drop-out, i.e. participants already more severely affected could be dropping out (in both groups admittedly at a similar rate), therefore shifting the group distributions (making the control group a less stable comparator and overestimating change in the intervention group).

As the paper presents baseline controlled analyses as a sensitivity and as long as one neither assumes differential drop out nor differential intervention effects, the impact on the cross-sectional pre-planned between-group comparisons should be minimal (under these assumptions we would still expect the same effect size). I nevertheless thought I just share this observation as it is a fairly typical problem in trials and no mention of this was found (see also (15) above).

Reply to R3#PO: We have intentionally refrained from commenting on these aspects due to the nature of the analysis we would be commenting on (exploratory, unplanned) and on the fact that our study was not calibrated to measure secular trends. Of course, we do agree with the referee that RCTs typically suffer from the problem of drop-outs, although this is a very general issue that applies to all of the literature of this kind (e.g. Bell et al., 2014, *BMC Med Res Methodol* 14, 118). In order to check the robustness of our results to a possible bias due to informative dropouts, we have included in our Supplementary materials the results of a linear mixed model (LMM), that simultaneously estimates the treatment effect on the outcome at all timepoints (T1 and T2, in our case) and produces unbiased treatment effect estimates in case of missingness at random (MAR; Ashbeck and Bell, 2016, *BMC Med Res Methodol* 16, 43). We controlled for values at T0 and, to make the MAR assumption plausible, we also included the binary variable on whether more than one log-in was performed, given its likely association with dropping out from measurements at both time points.

References

- Haushofer J., Shapiro J. (2016), The Short-term Impact of Unconditional Cash Transfers to the Poor: Experimental Evidence from Kenya, *The Quarterly Journal of Economics*, 131 (4), 1973–2042.
- Bell M.L., Fiero M., Horton N.J., Chiu-Hsieh Hsu (2014). Handling missing data in RCTs; a review of the top medical journals. *BMC Med Res Methodol* 14, 118.
- Ashbeck, E.L., Bell, M.L. (2016) Single time point comparisons in longitudinal randomized controlled trials: power and bias in the presence of missing data. *BMC Med Res Methodol* 16, 43.